# Myokine mediated muscle-kidney crosstalk suppresses metabolic reprogramming and fibrosis in damaged kidneys

Hui Peng[1,2], Qianqian Wang[1,2], Tanqi Lou[1], Jun Qin[3], Sungyun Jung[3], Vivekananda Shetty[4], Feng Li[4], Yanlin Wang[2], Xin-hua Feng[5], William E. Mitch[2], Brett H. Graham [iD] [6] & Zhaoyong Hu[2]

Kidney injury initiates metabolic reprogramming in tubule cells that contributes to the development of chronic kidney disease (CKD). Exercise has been associated with beneficial effects in patients with CKD. Here we show that the induction of a myokine, irisin, improves kidney energy metabolism and prevents kidney damage. In response to kidney injury, mice with muscle-specific PGC-1α overexpression (mPGC-1α) exhibit reduced kidney damage and fibrosis. Metabolomics analysis reveals increased ATP production and improved energy metabolism in injured kidneys from mPGC-1α mice. We identify irisin as a serum factor that mediates these metabolic effects during progressive kidney injury by inhibiting TGF-β type 1 receptor. Irisin depletion from serum blunts the induction of oxygen consumption rate observed in tubule cells treated with mPGC-1α serum. In mice, recombinant irisin administration attenuates kidney damage and fibrosis and improves kidney functions. We suggest that myokine-mediated muscle-kidney crosstalk can suppress metabolic reprograming and fibrogenesis during kidney disease.

[1] Nephrology Division, the Third Affiliated Hospital of Sun Yat-sen University, Guangzhou, China. [2] Nephrology Division, Department of Medicine, Baylor College of Medicine, Houston, TX, USA. [3] Department of Biochemistry and Molecular Biology, Baylor College of Medicine, Houston, TX, USA. [4] The Metabolomics and Proteomics Core, Baylor College of Medicine, Houston, TX, USA. [5] Department of Surgery, Baylor College of Medicine, Houston, TX, USA. [6] Department of Molecular and Human Genetics, Baylor College of Medicine, Houston, TX, USA. Correspondence and requests for materials should be addressed to Z.H. (email: zhaoyonh@bcm.edu)

Acute kidney injury (AKI) is a frequent complication of hospitalized patients, and it can progress to chronic kidney disease (CKD), thereby increasing a patient's risk of morbidity and mortality[1,2]. Unfortunately, there are no uniformly effective therapeutic interventions that prevent kidney tubule cell damage in AKI or CKD. However, there is evidence that physical exercise can slow the progression of chronic disorders[3], and clinical reports conclude that exercise can decrease the risk of developing progressive CKD[4]. If exercise can benefit the outcomes of patients with CKD, it can be speculated that skeletal muscle activity limits the degree of damage to kidney cells. The

presence of such crosstalk between skeletal muscles and kidneys is suggested because events in skeletal muscles can influence metabolic changes in other organs, such as adipose tissues or brain cells[5,6]. This type of crosstalk response has not been extended to determine whether communications between the skeletal muscles and the kidneys can suppress kidney damage.

Identifying whether other organs affect the severity of kidney tubule cell damage is a fertile area to explore because biochemical mechanisms can suppress ongoing cellular damage and function loss in organs[3]. In kidney, progressive tubule cell damage results in low ATP levels in cells due to defects in the oxidation of

substrates or other metabolic events[7]. This is relevant because correction of defective energy metabolism in kidney tubule cells can increase cellular levels of ATP, resulting in protection of mice from developing AKI[8]. Another biochemical response that benefits kidney tubule cell metabolism is the replacement of low levels of niacinamide[9]. Replacing cellular levels of nicotinamide adenine dinucleotide improves mitochondrial function and energy metabolism. Improvements in these factors were found to increase ATP production, counteract kidney damage, and suppress the development of renal fibrosis.

The pathogenesis of interstitial fibrosis occurring in injured kidneys involves induction of TGF-β1 expression and the development of inflammation, fibroblast activation, and extracellular matrix deposition[7,10,11]. As a key mediator of kidney fibrosis, TGF-β1 not only activates the expression of fibrotic genes but also stimulates Warburg-like metabolic reprogramming in kidney cells[8,12]. The latter response is relevant because metabolic reprogramming in kidney cells is present during kidney injury and contributes to the pathogenesis of renal fibrosis[13]. Despite reports that certain biochemical responses can suppress the severity of kidney cell injury, we have found no crosstalk mechanisms that originate in non-kidney organs and prevent progressive kidney cell injury.

Because muscle exercise can benefit the outcomes of CKD patients, we investigated how a substitute for exercise, overexpression of PGC-1α only in the skeletal muscles (mPGC-1α)[14], affects recovery from kidney tubule cell damage in three well-established mouse renal injury models. We found that the development of kidney interstitial fibrosis is suppressed in mPGC-1α mice with prevented metabolic reprograming in injured tubule cells. We also identified a myokine, irisin, mediating these beneficial responses in mPGC-1α mice. Our results suggest that muscle-kidney crosstalk can ameliorate tubule cell damage and kidney fibrosis.

## Results

**mPGC-1α limits fibrosis in damaged kidneys**. To begin testing our hypothesis that muscle-kidney crosstalk suppresses kidney tubule damage and the development of fibrosis, we initially examined renal fibrogenesis in mPGC-1α mice that had been treated with folic acid. Two weeks after folic acid injection, the mPGC-1α mice not only exhibited reduced tubular dilatation and cellular damage; their kidneys also had significantly decreased interstitial fibrosis in comparison to those of littermate, wild type mice that were treated in the same way (Fig. 1a, b). Next, we used a PCR-based messenger RNA (mRNA) array to assess the activation of genes involved in matrix remodeling, inflammation, and

TGF-β1 signaling (Fig. 1c). In the kidneys of wild type and mPGC-1α mice that had not been treated with folic acid, there were comparable expressions of genes (Fig. 1d, upper panel). However, at 2 weeks after folic acid treatment, expression of 37 out of 84 genes had changed (either up-regulated or down-regulated) in the damaged kidneys of wild type mice in comparison to results from mice with uninjured kidneys (Fig. 1d, middle panel). In contrast, only 12 genes changed in the kidneys of mPGC-1α mice treated with folic acid in comparison to the results from uninjured mPGC-1α mice (Fig. 1d, lower panel). Thus, 25 genes were corrected in the kidneys from mPGC-1α mice treated with folic acid. A KEGG Pathway analysis revealed that these 25 genes were closely related to TGF-β1 signaling and the interactions of receptors and cytokines (Fig. 1e).

Using quantitative, real-time PCR, we confirmed mRNA expression of key fibrotic genes, including fibronectin (FN), collagen 1A (COL1A), connective tissue growth factor (CTGF), and α-smooth muscle actin (α-SMA). The expressions of these genes were significantly suppressed in the kidneys of mPGC-1α mice after folic acid treatment compared to the results from wild type mice treated identically (Fig. 1f). Concomitantly, protein levels of FN, COL1A, and α-SMA were markedly decreased in the kidneys of mPGC-1α mice treated with folic acid (Fig. 1g).

To assess whether the reduced kidney fibrosis was associated with improved kidney function, we examined serum creatinine levels and found that they were significantly lower in mPGC-1α mice treated with folic acid than in wild type mice treated similarly (Fig. 1h). In mice with UUO, the mPGC-1α mice exhibited lower levels of fibrotic proteins and less severe kidney damage than wild type mice with UUO (Supplementary Fig. 1).

As the third model of kidney damage, we examined mice at 3 months after subtotal nephrectomy (CKD). We observed severe glomerulosclerosis, tubular atrophy, and interstitial fibrosis in the kidneys of wild type mice with CKD; these pathological changes were significantly less severe in the kidneys of mPGC-1α mice with CKD (Supplementary Fig. 2). Notably, kidney function, which was evaluated by serum creatinine and urine albumin-to-creatinine ratios, was also significantly better in mPGC-1α mice with CKD (Supplementary Fig. 2e). Taken together, the results indicate that mPGC-1α attenuated kidney fibrogenesis and function loss after kidney injury, suggesting crosstalk between the skeletal muscles and the kidneys.

**mPGC-1α prevents tubule damage and TGF-β1 activation**. To ascertain the mechanism by which mPGC-1α limits fibrogenesis after kidney injury, we examined kidney tubule cells in the early stages after folic acid injury. At 7 days after folic acid injection, we

**Fig. 1** Overexpression of PGC-1α in skeletal muscles suppresses fibrogenesis in Folic Acid Nephropathy (FAN). **a** PAS staining of kidney sections from wild type (Wt), muscle-specific PGC-1α overexpression (mPGC-1α) with or without folic acid-induce nephropathy (FAN) are shown, representative image derived from 7 animals of each group. At 2 weeks after FAN, there were less tubule dilation, atrophy and brush-board loss in mPGC-1α mice vs. Wt mice with FAN. Scale bars, 50 μm. **b** Sirius red staining revealed there are less collagen deposition in kidneys from FAN-mPGC-1α mice. Quantitative analysis of renal interstitial collagen in different groups as indicated. $^{**}P < 0.01$ for Wt-FAN vs. Wt-Sham, $^{*}P < 0.05$ mPGC-1α -FAN vs. mPGC-1α -Sham, and $^{#}P < 0.05$ for mPGC-1α -FAN vs. Wt-FAN, one-way ANOVA with Bonferroni's multiple comparison test. Data were presented as mean ± s.e.m.; $n = 7$ per group. Scale bars, 100 μm. **c** Heat map of RT-PCR array for genes related to fibrosis with hierarchical clustering analysis ($n = 3$). **d** Statistical analysis of fibrosis PCR Array as volcano plots (statistically significant gene expression differences were defined by $P < 0.05$ with Student–Newman–Keul's two-tailed, unpaired test). **e** KEGG pathway analysis of limited genes by mPGC-1α indicated that TGF-β signaling is one of the most affected pathway in kidneys of mPGC-1α mice (Arrow). **f** the mRNA level of fibrotic genes in kidneys from each group was confirmed using Real time RT- quantitative PCR. Data were presented as mean ± s.e.m.; $^{**}P < 0.01$ for Wt-FAN vs. Wt-Sham, $^{#}P < 0.05$ for mPGC-1α-FAN vs. Wt-FAN, one-way ANOVA with Bonferroni's multiple comparison test, $n = 6$ per group. **g** Fibrotic proteins were examined by western blotting. Quantitative measurements of Fibronectin (FN), Collagen 1A and α-SMA protein expression in kidneys of Wt and mPGC-1α mice with or without FAN. $^{**}P < 0.01$ for Wt-FAN vs. Wt-Sham and $^{#}P < 0.05$ for mPGC-1α-FAN vs. Wt-FAN, one-way ANOVA with Bonferroni's multiple comparison test, $n = 6$ per group. **h** Kidney function was assessed using serum creatinine concentration in each group. $^{**}P < 0.01$ Wt-FAN vs. Wt-Sham, $^{*}P < 0.05$ for mPGC-1α-FAN vs. mPGC-1α-Sham, and $^{#}P < 0.05$ for mPGC-1α-FAN vs. Wt-FAN, $n = 7$ per group, one-way ANOVA with Bonferroni's multiple comparison test

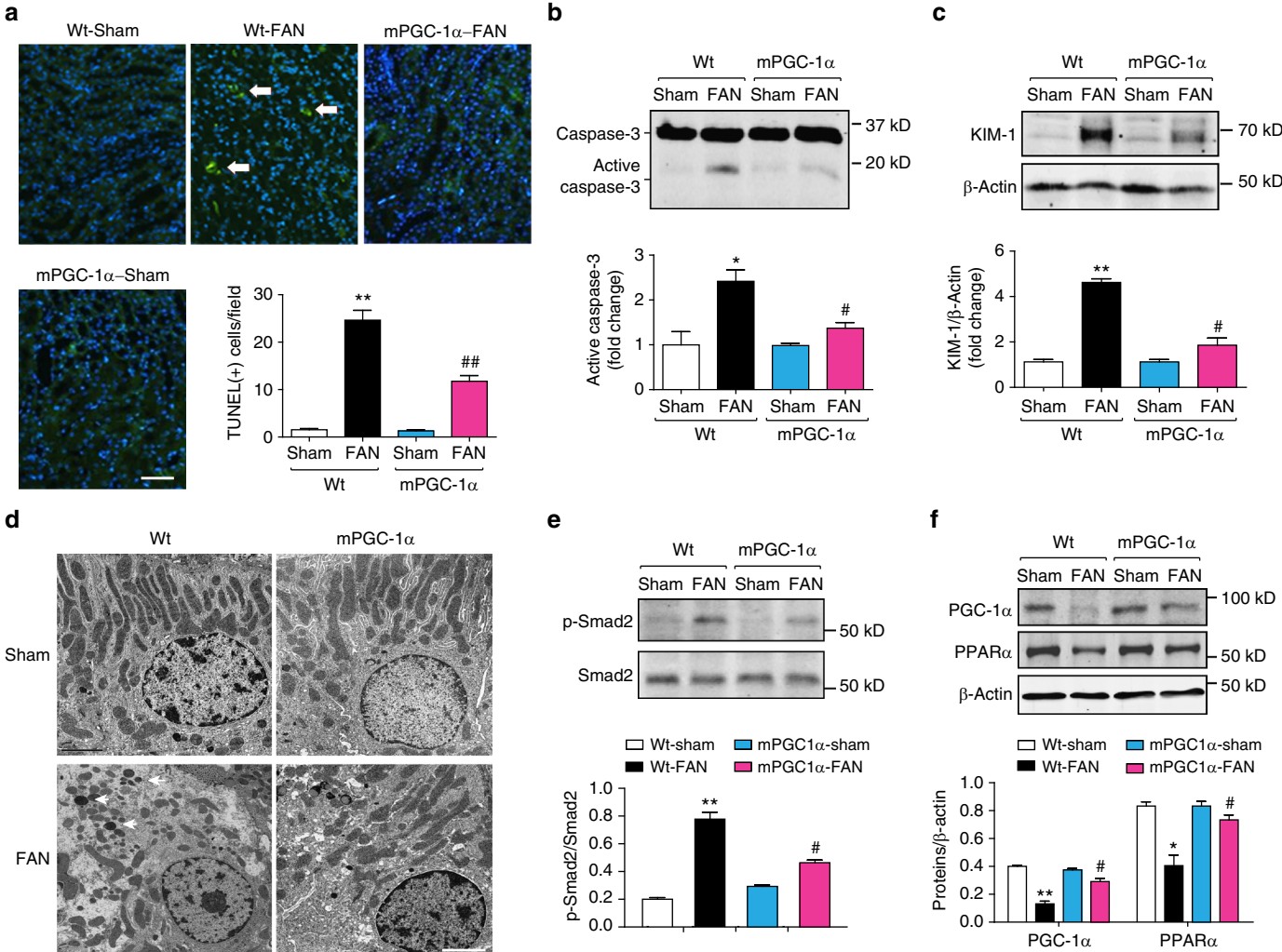

**Fig. 2** mPGC-1α prevents tubule cells damage and activation of TGF-β1 signaling in kidneys of mice with FAN for 7 days. **a** Representative images of TUNEL assay revealed an increase in apoptosis in tubules from Wt mice with FAN, this marker of tubule cells damage significantly reduced in kidneys from mPGC-1α mice treated with folic acid. Data were presented as mean ± s.e.m.; ${}^{**}P < 0.01$ for Wt-FAN vs. Wt-Sham and ${}^{##}P < 0.01$ for mPGC-1α-FAN vs. Wt-FAN ($n = 5$ per group, one-way ANOVA with Bonferroni's multiple comparison test). Scale bars, 100 μm. **b** Caspase-3 activation (cleaved caspase-3) were evaluated with western blots. Quantitative measurements indicated an increase in cleaved caspase-3 in kidney from Wt mice, and this response was blunted in kidneys of mPGC-1α mice despite of FAN. ${}^{*}P < 0.05$ Wt-FAN vs. Wt-Sham and ${}^{#}P < 0.05$ for mPGC-1α-FAN vs. Wt-FAN ($n = 5$ per group, one-way ANOVA with Bonferroni's multiple comparison test). **c** Western blot of Kidney Injury Marker 1 (KIM-1), its expression was significantly suppressed in kidneys from mPGC-1α mice with FAN. ${}^{**}P < 0.05$ for Wt-FAN vs. Wt-Sham and ${}^{#}P < 0.05$ for mPGC-1α-FAN vs. Wt-FAN ($n = 5$ per group, one-way ANOVA with Bonferroni's multiple comparison test). **d** Electron microscopy showed that fragmentation of mitochondria and "protein droplets" in proximal tubule cells of Wt mice with FAN (Arrows), these changes were almost eliminated in the tubular cells of mPGC-1α mice with FAN. $n = 5$ per group, Scale bars, 2.5 μm. **e** Western blots of p-Smad2 in WT and PGC-1α mice with or without FAN. ${}^{**}P < 0.05$ for Wt-FAN vs. Wt-Sham. ${}^{#}P < 0.05$ for mPGC-1α-FAN vs. Wt-FAN ($n = 5$ per group, one-way ANOVA with Bonferroni's multiple comparison test). **f** The expression of PPARα and its co-activator (PGC-1α) were examined by western blot. Data were presented as mean ± s.e.m.; ${}^{*}P < 0.05$ or ${}^{**}P < 0.01$ for Wt-FAN vs. Wt Sham and ${}^{#}P < 0.05$ for mPGC-1α-FAN vs. WtFAN, $n = 5$ per group (one-way ANOVA with Bonferroni's multiple comparison test)

evaluated apoptosis in kidney tubule cells using a TUNEL assay. Apoptosis rates were significantly lower in mPGC-1α mice than in wild type mice (Fig. 2a). Likewise, levels of activated caspase-3 in injured kidneys were significantly suppressed in mPGC-1α mice vs. the result from wild type mice (Fig. 2b). We also found that the expression of Kidney Injury Marker-1 (KIM-1), was lower in the folic acid-damaged kidneys of mPGC-1α mice than in wild type mice (Fig. 2c)[15].

In mPGC-1α mice damaged with folic acid, electron microscopy revealed only slight mitochondrial fragmentation into short rods or spheres in kidney tubule cells, whereas more extensive injury was revealed in the cells of wild type mice treated identically (Fig. 2d). As another sign of suppressed kidney tubule

cell damage, we found significantly fewer 'protein droplets' in tubule cells from folic acid-treated mPGC-1α mice than in identically treated wild type mice (Fig. 2d, arrows). Because activated TGF-β1signaling can suppress peroxisome proliferator-activated receptors α (PPARα) and PGC-1α, resulting in impaired energy metabolism in kidney tubule cells[8,12,16], we evaluated TGF-β1 signaling in kidneys from wild type or mPGC-1α mice that were treated with folic acid. As expected, there was an increase in Smad2/3 phosphorylation in damaged kidneys from wild type mice, and this response was significantly suppressed in the kidneys of mPGC-1α mice (Fig. 2e). Concomitantly, the expression of PPARα and PGC-1α were significantly suppressed in folic acid-damaged kidneys from wild type mice but not in

damaged kidneys from mPGC-1α mice (Fig. 2f). Thus, PGC-1α overexpression in muscles protects kidney tubule cells from mitochondria damage and progressive injury.

**mPGC-1α corrects metabolic reprogramming in injured kidneys.** We measured mitochondrial respiration in freshly isolated kidney tubules from wild type and mPGC-1α mice. In uninjured

kidney tubules, basal oxygen consumption rates (OCR) of wild type and mPGC-1α mice did not differ significantly (Fig. 3a, c). However, at 7 days after folic acid treatment, mitochondrial respiration decreased significantly in the tubules of wild type mice, and this abnormality was largely prevented in the tubules of mPGC-1α mice (Fig. 3b, c). After folic acid injury to the tubules of wild type mice, the rates of maximal respiration and spare

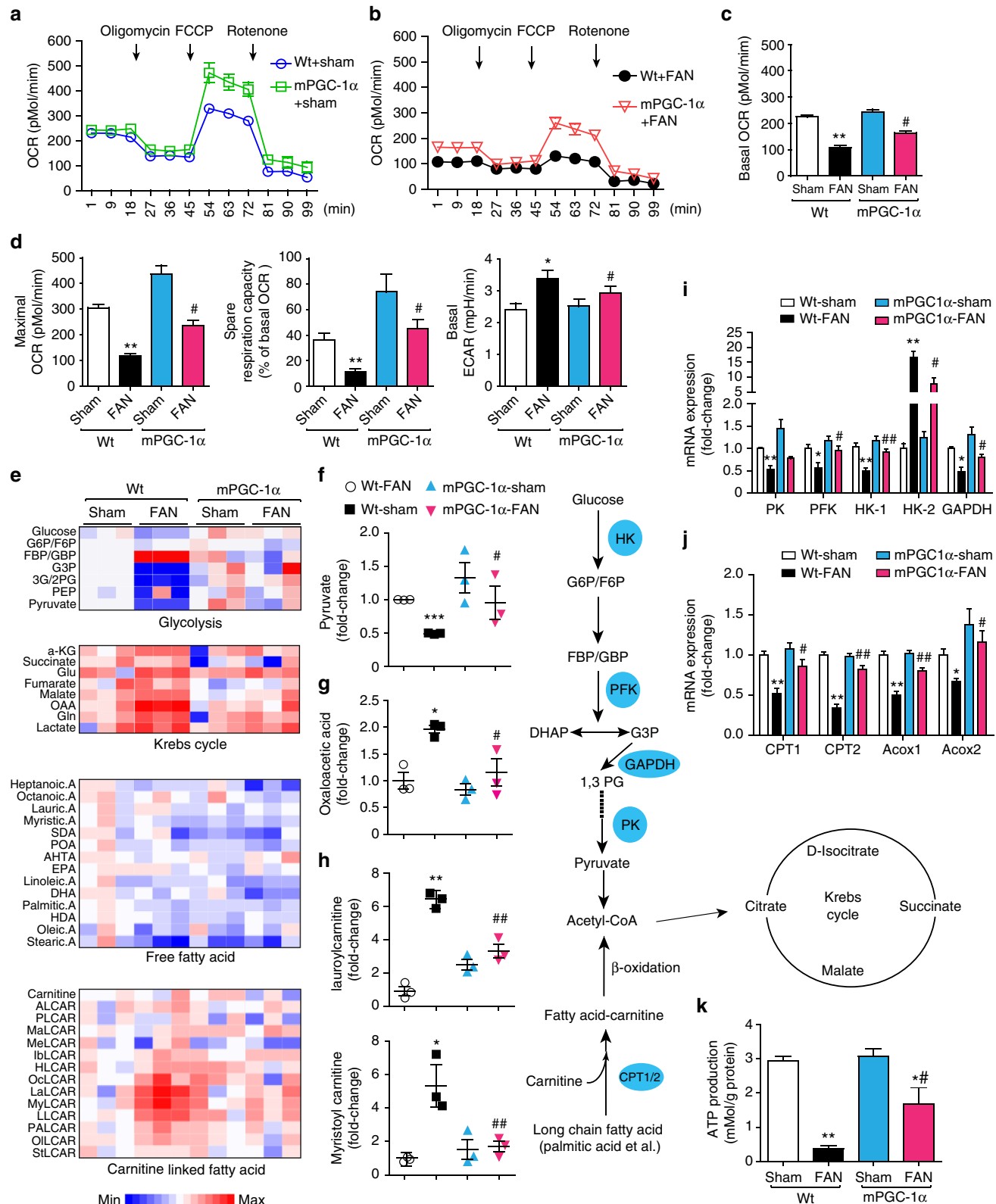

respiratory capacity were lower than those in uninjured tubules. Notably, these defects were significantly improved in the tubules of mPGC-1α mice despite folic acid injury (Fig. 3d, left and middle panel). The extracellular acidification rate (ECAR) in tubules from folic acid-treated mPGC-1α mice also was decreased vs. the results from injured tubules of wild type mice (Fig. 3d, right panel). In the kidney tubules of mice with 7 days of UUO, there were also lower values of oxygen consumption and greater acidification than in the tubules of uninjured, wild type mice. These negative responses were partially blocked in the tubules of mPGC-1α mice (Supplementary Fig. 3a and b). Thus, mPGC-1α improves energy metabolism and mitochondrial function in damaged kidney tubule cells.

To understand how tubule cell energy metabolism was improved in mPGC-1α mice, we used metabolomic profiling to examine how metabolites change following folic acid injury (Fig. 3e). In damaged kidneys of wild type mice, glucose metabolites, including pyruvate, were substantially reduced, indicating impaired glucose metabolism. But, in the injured kidneys of mPGC-1α mice, glucose metabolites were significantly increased (Fig. 3f). In the injured kidneys of wild type mice, there was also marked accumulation of Krebs Cycle metabolites (e.g., malate, oxaloacetic, and glutamine), demonstrating that folic acid impairs the metabolism in mitochondria (Fig. 3g). These defects were also substantially improved in the kidneys of mPGC-1α mice. In addition, folic acid treatment of wild type mice led to the accumulation of carnitine-conjugated long chain fatty acids (e.g., lauroylcarnitine, myristoylcarnitine, and linoleoylcarnitine), indicating the presence of defects in fatty acid metabolism. In contrast, these abnormalities were largely corrected in injured kidneys from mPGC-1α mice (Fig. 3h).

To understand how these changes in metabolites might occur, we examined the mRNA of key enzymes involved in glucose metabolism (i.e., PFK and GAPDH) or fatty acid oxidation (i.e., CPT1 and CPT2). The expression of these enzymes was significantly reduced in the kidneys of wild type mice given folic acid, and, importantly, defects in the expression of these enzymes were significantly improved in the tubules of mPGC-1α mice (Fig. 3i, j). Notably, only the mRNA of hexokinase 2 (HK2) was elevated in the damaged kidneys of wild type mice. This result could reflect an increase in the rate of glycolysis[17]. Finally, we measured the intracellular ATP level in damaged kidneys. It was significantly higher in the injured kidneys of mPGC-1α mice than in the kidneys of wild type mice (Fig. 3k). In wild type mice with 7 days of UUO, there were lower expressions of key enzymes involved in glucose metabolism or fatty acid oxidation in kidney tubule cells; expectedly, these defects were significantly corrected in the kidneys from mPGC-1α mice with 7 days of UUO (Supplementary Fig. 3c and d). These improvements significantly increased ATP production in the damaged kidneys of mPGC-1α mice (Supplementary Fig. 3e). Thus, mPGC-1α prevents metabolic reprogramming and improves ATP levels in injured kidneys.

## Myokines stimulate mitochondrial functions in tubule cells.

mPGC-1α mice can increase the expression of several myokines in muscle[18]. To identify the serum factor that might influence energy metabolism in kidney tubule cells, we initially measured the mRNAs of 24 myokines present in the tibialis anterior muscles from wild type and mPGC-1α mice. Six myokine genes (irisin, BDNF, IL-15, ANGPTL4, FGF-21, and CTSB) were markedly upregulated in the muscles of mPGC-1α mice (Fig. 4a). We also used ELISA to assess the serum levels of irisin, BNDF, and IL-15, and the measurements confirmed that levels of these three myokines were significantly higher in mPGC-1α mice than in wild type mice (Fig. 4b). To determine whether the myokines improved mitochondrial respiration in kidney tubule cells, we treated primary cultures of kidney tubule cells with serum from wild type or mPGC-1α mice. The basal level of cellular respiration (OCR) was affected minimally by exposure to either type of serum. In contrast, serum from mPGC-1α mice significantly increased the maximal respiratory capacity and ATP-coupled respiration in comparison to results from cells treated with wild type serum (Fig. 4c). To identify specific myokines in serum from mPGC-1α mice that improve mitochondrial function, we divided serum samples from wild type or mPGC-1α mice into 4 fractions (SF): SF1 > 100 kD, SF2 is between 100 and 50 kD, SF3 is between 50 and 10 kD, and SF4 < 10 kD. Each fraction was incubated with a primary culture of kidney tubule cells; only the SF3 fraction from mPGC-1α mice significantly stimulated ATP-coupled respiration and maximal respiratory capacity (Fig. 4d). Responses to the SF3 fraction were eliminated by heating it to 95 °C for 10 min (Supplement Fig. 4a). Using mass spectrometry, we confirmed that irisin is present in the SF3 serum fraction from mPGC-1α mice (Fig. 4e). Notably, treatment with an anti-irisin antibody to remove irisin revealed that the SF3 fraction no longer improved mitochondrial respiration (Fig. 4f). We conclude that irisin, presented in serum from mPGC-1α mice, improves mitochondrial respiration and energy metabolism in damaged tubule cells.

## Irisin counteracts TGF-β1-induced metabolic reprogramming.

We measured mitochondrial respiration in primary cultures of tubule cells after exposure to recombinant irisin (2 μg ml⁻¹) for 16 h. Irisin increased mitochondrial oxygen consumption and maximal respiratory capacity, as well as ATP-coupled respiration, in comparison to the results found in untreated tubular cells (Fig. 5a). To determine whether irisin-stimulation of aerobic metabolism utilizes glucose and/or fatty acids as substrates, we measured oxygen consumption in tubule cells that were pretreated with irisin and then exposed to glucose or palmitate. As

**Fig. 3** mPGC-1α suppresses metabolic reprogramming in injured kidney after 7 days FAN. **a** Cellular respiration was examined in fresh isolated tubules from kidneys of Wt sham and mPGC-1α sham mice by measuring Oxygen Consumption Rate (OCR) and Extracellular Acidification Rate (ECAR) using XFe24 extracellular flux analyzer. **b** Cellular respirations in fresh isolated tubules form Wt and mPGC-1α mice with FAN **c** Measurements of basal OCR in each group as indicated. **d** Measurements of Maximal respiration and Spare respiration rate (percentage of basal respiration), and Basal ECAR in each group. Data (**c**, **d**) was presented as mean ± s.e.m.; *$P < 0.05$ or **$P < 0.01$ for Wt-FAN vs. Wt-Sham and #$P < 0.05$ for mPGC-1α-FAN vs. Wt-FAN, $n = 5$ per group (one-way ANOVA with Bonferroni's multiple comparison test). **e** Metabolomics analysis of kidney cortex form each group as indicated. Heat map shows relative levels of metabolites in kidneys from mice each group ($n = 3$). **f–h** Statistical analysis of representative metabolites in glycolysis pathway, fatty acid metabolism and Krebs cycle. *$P < 0.05$ or **$P < 0.01$ or ***$P < 0.001$ for Wt-FAN vs. Wt-Sham. #$P < 0.05$ for mPGC-1α-FAN vs. Wt-FAN ($n = 3$ per group, one-way ANOVA with Bonferroni's multiple comparison test). **i** and **j** Real-time qPCR analysis of enzymes expression in glycolysis pathway or Fatty Acid oxidation pathway. *$P < 0.05$ or **$P < 0.01$ for Wt-FAN vs. Wt-Sham. # $P < 0.05$ or ##$P < 0.01$ for mPGC-1α-FAN vs. Wt-FAN ($n = 5$ per group, one-way ANOVA with Bonferroni's multiple comparison test). **k** ATP concentration in kidney cortex tissue from normal mice or mice treated with FAN. Data were presented as mean ± s.e.m.; **$P < 0.01$ for Wt-FAN vs. Wt-Sham. *$P < 0.05$ for mPGC-1α-FAN vs. mPGC-1α-Sham. #$P < 0.05$ for mPGC-1α-FAN vs. Wt-FAN ($n = 5$ per group, one-way ANOVA with Bonferroni's multiple comparison test)

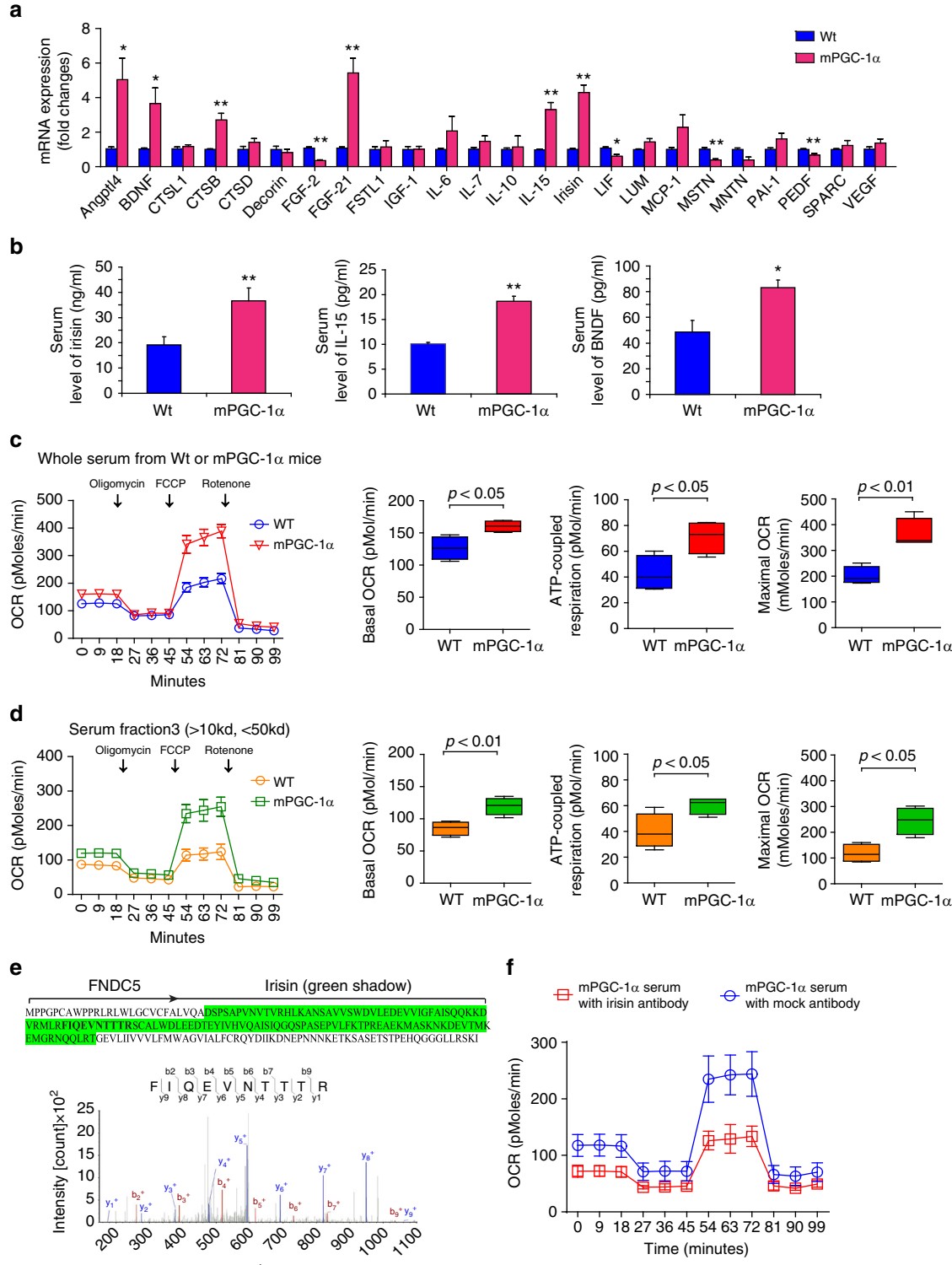

**Fig. 4** Myokines in serum from mPGC-1α mice stimulate tubular cell respiration. **a** Real-time qPCR analysis of 24 known myokines expression in skeletal muscles of Wt and mPGC-1α mice. *$P < 0.05$ or **$P < 0.01$ for mPGC-1α vs. Wt mice ($n = 3$ per group, Student-Newman-Kuel's two-tailed, unpaired tests). **b** Serum concentration of selected myokines (irisin, iL-15 and BNDF) in Wt and PGC-1a mice. Data were presented as mean ± s.e.m.; *$P < 0.05$ or **$P < 0.01$ for mPGC-1α vs. Wt mice ($n = 5$ per group; Student-Newman-Kuel's two-tailed, unpaired tests). **c** Primary tubule cell cultures were incubated with 10% of serum form WT or PGC-1α mice for 12 h followed by cellular respiration measurement. Measurements of basal OCR, ATP-coupled respiration and maximal respiration in each group ($n = 5$ per group, Student-Newman-Kuel's two-tailed, unpaired tests). **d** Primary tubule cell cultures were treated with different serum fractions, only serum fraction 3 (<50 kd, but > 10 kd) exhibited a significant increase in cell respiration ($n = 5$ per group, Student-Newman-Kuel's two-tailed, unpaired tests). **e** In serum fraction 3, the present of irisin was confirmed by mass spectra. **f** Sera from mPGC-1α mice was incubated anti-irisin antibody or mock antibody for 4 h and following with an incubation with protein A/G for 1 h at 4 °C. After spindown to remove irisin and lgG, the sera were used to treat primary cultured tubule cells following a measurement of cellular respiration

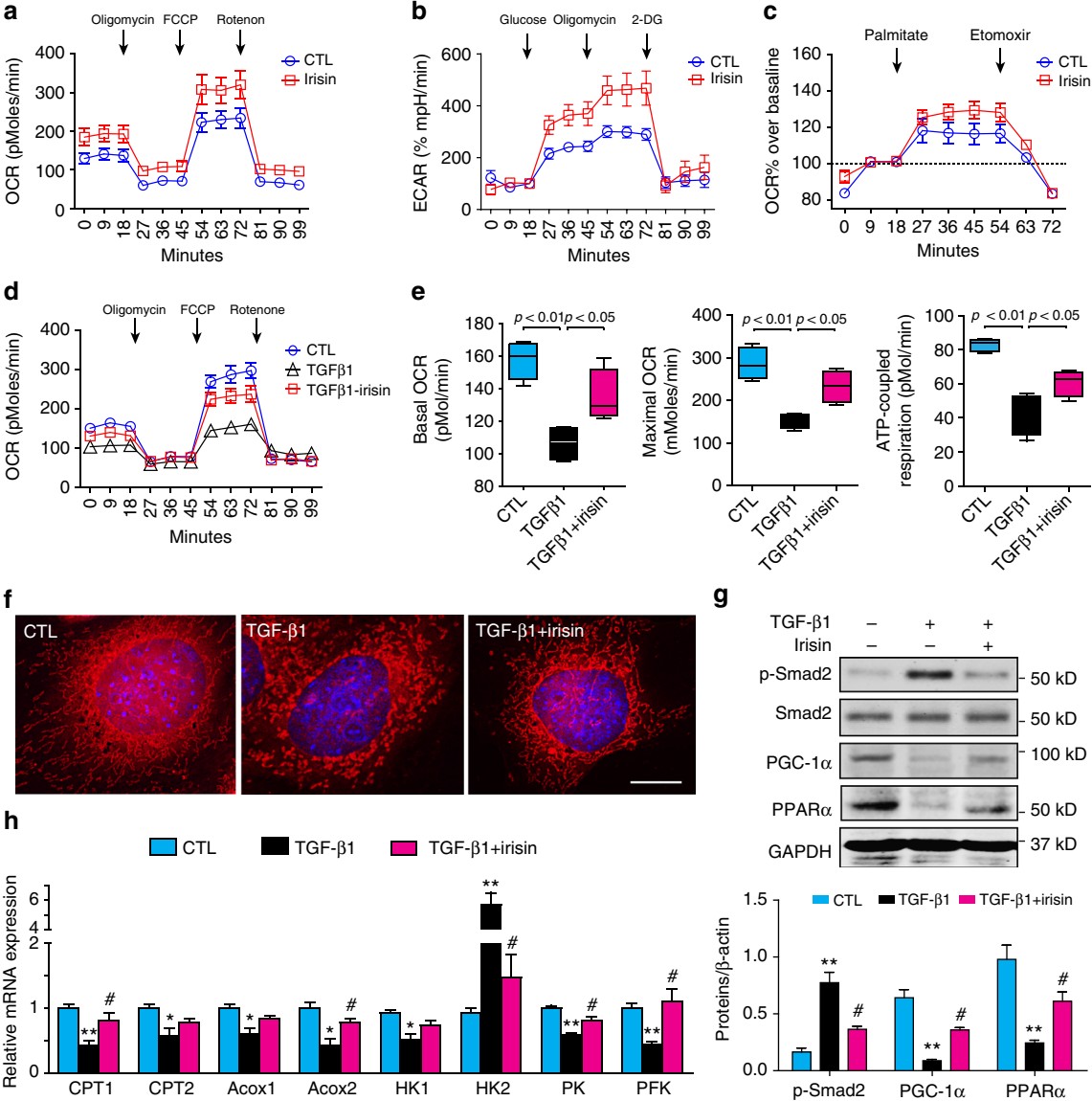

**Fig. 5** Recombinant irisin counteracts TGF-β1-induced metabolic reprogramming in primary tubule cells. **a–c** Primary tubule cells cultures were treated with recombinant irisin (2 µg ml$^{-1}$) for 12 h, cell respiration was examined with mito-stress assay **a**, glycolysis assay **b** and Fatty acid assay **c**. CTL: control, with no addition of irisin. **d, e** Cell respiration of tubule cells treated with TGF-β1(2 ng ml$^{-1}$, for 12 h) or pretreated with irisin before adding TGF-β1. Statistical analysis of basal OCR, maximal respiration and ATP-coupled respiration ($n = 5$ per group, one-way ANOVA with Bonferroni's multiple comparison test). **f** Mitochondria morphology was examined with mito-tracker staining after tubule cells treated with TGF-β1 or TGF-β1 plus irisin. Scale bars, 10 µm. **g** Western blot showing the expression of Smad2, PPARα and PGC-1α in tubule cells treated with TGF-β1 or TGF-β1 plus irisin. Quantitative measurements of Smad2, PPARα and PGC-1α protein expression in each group as indicated. $^{**}P < 0.01$ for TGF-β1 vs. CTL, $^{\#}P < 0.05$ for TGF-β1 + irisin vs. TGF-β1 treated group ($n = 5$ per group, one-way ANOVA with Bonferroni's multiple comparison test). **h** Real-time qPCR showing irisin counteracts TGF-β1-induced suppression of enzymes in energy metabolic pathways. Data were presented as mean ± s.e.m.; $^{*}P < 0.05$ or $^{**}P < 0.01$ for TGF-β1 vs. CTL. $^{\#}P < 0.05$ for TGF-β1 + irisin vs. TGF-β1 treated group, $n = 5$ per group (one-way ANOVA with Bonferroni's multiple comparison test)

shown in Fig. 5b, irisin stimulated both maximal respiration and extracellular acidification after adding glucose; this response was eliminated by adding 2-deoxyglucose. Irisin significantly increased tubule cell oxygen consumption when palmitate was provided, and this reaction was halted by adding Etomoxir, an inhibitor of fatty acid β-oxidation (Fig. 5c). Thus, irisin stimulates aerobic metabolism of both glucose and fatty acids in kidney tubule cells.

Next, we examined whether irisin counteracts TGF-β1-induced metabolic reprograming in kidney tubule cells. As shown in Fig. 5d, adding TGF-β1 significantly decreased basal, maximal and ATP-coupled oxygen consumption in kidney tubule cells,

and irisin reversed these responses (Fig. 5e). The decrease in the ATP-coupled oxygen consumption rate was associated with mitochondrial swelling and fragmentation, and irisin treatment prevented this negative response (Fig. 5f). Because TGF-β1 can suppress PPARα and PGC-1α in kidney tubule cells, we examined its mediator: Smad2/3 phosphorylation. As shown in Fig. 5g irisin significantly suppressed Smad2/3 phosphorylation and increased PPARα and PGC-1α. Consistent with these improvements, the expressions of key enzymes mediating aerobic glycolysis and β-oxidation were also preserved by irisin treatment (Fig. 5h). Thus, irisin counteracts TGF-β1-induced metabolic reprogramming in kidney tubule cells.

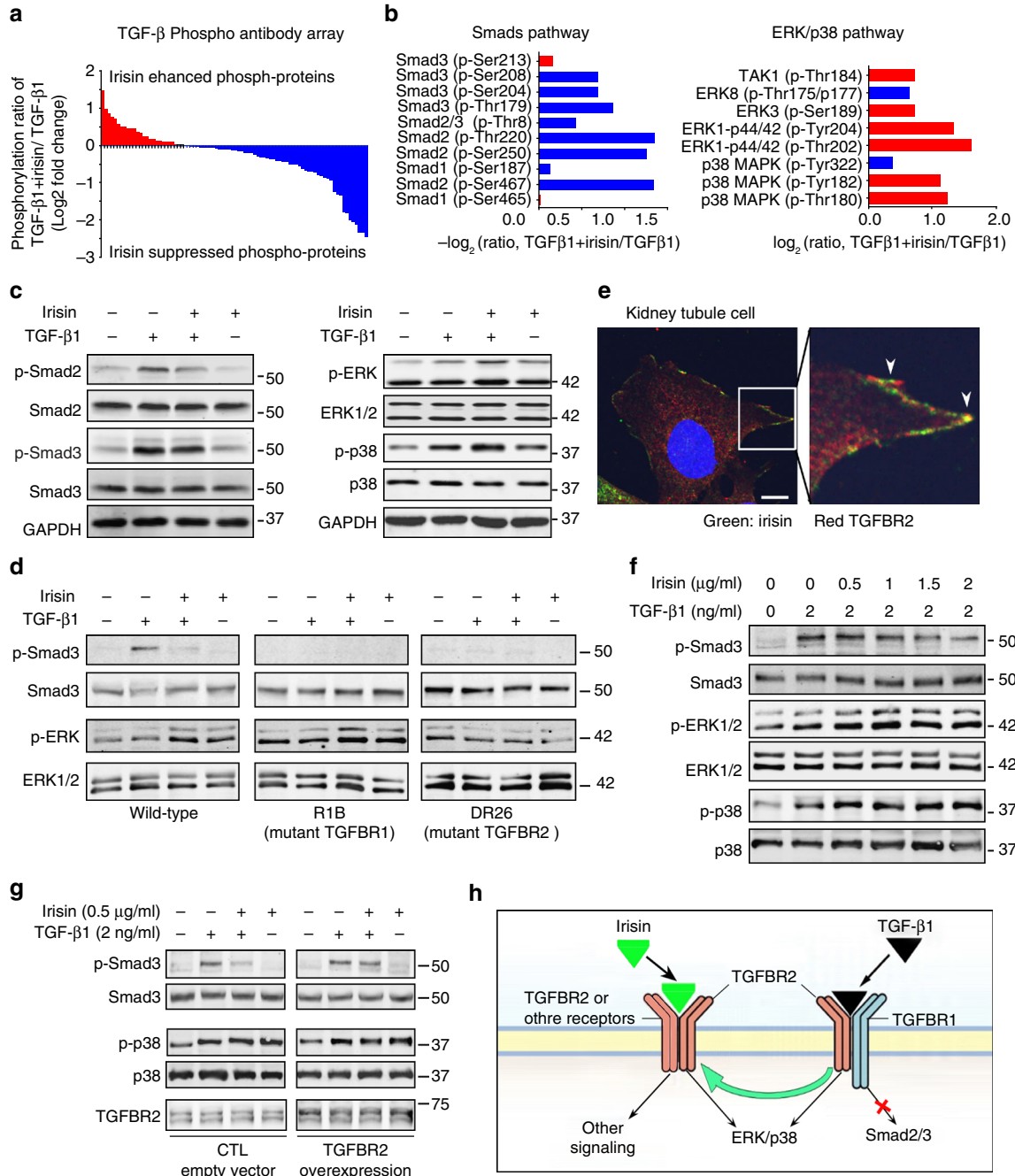

**Fig. 6** Irisin antagonizes TGF-β1 via interfering TGFBR 1 activation in primary tubule cell cultures. **a** TGF-β1 Signaling Phosphoproteome array in tubule cells (TGF-β1 vs. TGF-β1 plus irisin). Data showing the fold change of enhanced phosphoproteins (red) and suppressed phosphoproteins (blue). **b** Irisin enhanced ERK/p38 signaling but suppressed Smads signaling upon TGF-β1 stimulation. **c** Representative western blots (from three independent experiments) confirmed that irisin suppressed Smad2/3 phosphorylation (left panel) while stimulated ERK and p38 phosphorylation. **d** Representative western blots revealing irisin-induced ERK1/2 phosphorylation was eliminated in DR 26 (TGFBR2 mutant) but not in R1B cell (TGFBR1 mutant). Results were repeated in three independent experiments. **e** In primary cultured tubule cell, fluorescein labeled irisin (green) bound to cell membrane that was co-localized with TGFBR2 (red). **f** Representative western blot showing irisin dose-dependently suppressing TGF-β1-induced Smad2/3 phosphorylation. Results were repeated in 3 independent experiments. **g** In tubule cells, overexpression of TGFBR2 blunted the inhibitory effect of irisin on Smad2/3 phosphorylation (representative western blot from three independent experiments). **h** A possible working model exhibiting how irisin compete TGF-β1 binding with TGFBR2

**Irisin inhibits the activation of TGF-β type 1 receptor**. To ascertain the mechanism by which irisin might counteract TGF-β1-induced metabolic reprograming in damaged kidney tubule cells, we preincubated tubule cells with irisin for 1 h and then added TGF-β1 for 2 h. Using a TGF-β-phospho-antibody array, we examined 86 phosphorylated proteins that are related to TGF-β signaling (Fig. 6a and Supplementary Table 1). In this preliminarily screen, we found that TGF-β1 increased the phosphorylation of Smad2/3; this increase was blunted by irisin (Fig. 6b left panel). Interestingly, TGF-β1-induced phosphorylation of ERK and p38 can be enhanced by irisin (Fig. 6b, right panel). Using western blotting, we confirmed that irisin

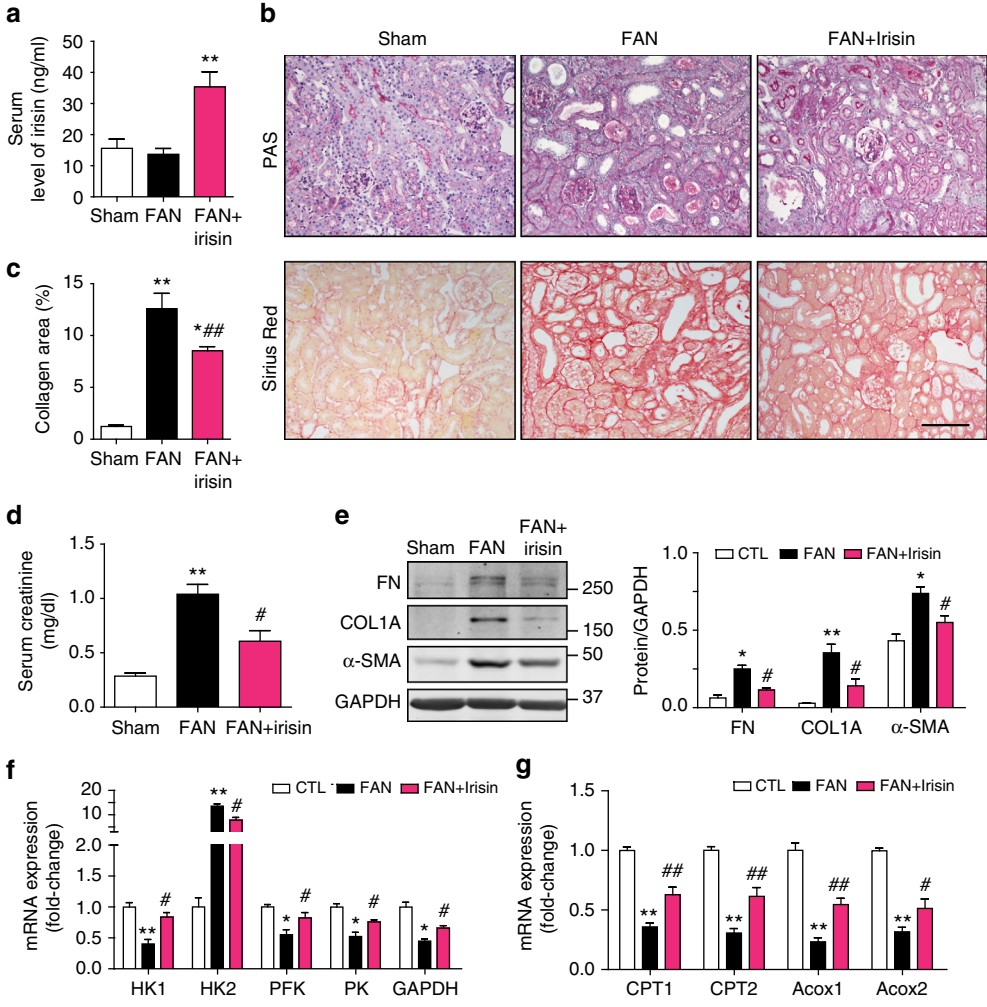

**Fig. 7** Recombinant irisin administration ameliorated renal fibrosis in FAN mice. **a** Irisin injection (200 μg kg⁻¹, ip daily for 2 weeks) achieved > 2 folds increase in serum (**P < 0.01 for FAN + irisin vs. Sham or FAN group, n = 5 per group, one-way ANOVA with Bonferroni's multiple comparison test). **b** Representative images of PAS staining showing less tubule dilation, atrophy and brush-board loss in FAN mice treated with recombinant irisin. **c** Sirius red staining revealing collagen deposition decreased in FAN mice treated with irisin. Quantitative analysis of interstitial collagen in different groups as indicated. **P < 0.01 for FAN vs. Sham, *P < 0.05 for FAN + irisin vs. FAN, and ##P < 0.01 for FAN + irisin vs. Sham. Data was presented as mean ± s.e.m. (n = 5 per group, one-way ANOVA with Bonferroni's multiple comparison test). Scale bars, 100 μm. **d** Kidney function (serum creatinine) was assessed in FAN mice after irisin treatment. **e**: Western blots showing fibrotic protein expression in kidneys from FAN mice treated with irisin. Quantitative measurements of Fibronectin (FN), Collagen 1A and α-SMA protein expression in kidneys from each group as indicated. *P < 0.05 or **P < 0.01 for FAN vs. Sham and #P < 0.05 for FAN + irisin vs. FAN (n = 5 per group, one-way ANOVA with Bonferroni's multiple comparison test). **f** and **g** Real-time qPCR analysis of enzymes expression in glycolysis **f** and fatty acid oxidation **g** pathway. Data were presented as mean ± s.e.m.; *P < 0.05 or **P < 0.01 for FAN vs. Sham and #P < 0.05 for FAN + irisin vs. FAN (n = 5 per group, one-way ANOVA with Bonferroni's multiple comparison test)

suppressed TGF-β 1-stimulated phosphorylation of Smad2/3 in tubule cells (Fig. 6c, left panel). We also confirmed that irisin intensified TGF-β1-induced phosphorylation of ERK and p38 (Fig. 6c, right panel). On the basis of these results, we speculate that irisin interacts with TGFBR2—synergistically stimulating ERK signaling—and that it simultaneously inhibits the recruitment of TGFBR1 to TGFBR2, therefore impeding Smad2/3 phosphorylation.

TGF-β1 interacts with the TGF-β type-2 receptor (TGFBR2) to recruit and activate the TGF-β type-1 receptor (TGFBR1)[19]. Active TGFBR1 phosphorylates and activates canonical, Smad2/3-medicated signaling[19], while TGFBR2 activates signaling pathways, including TAK1, ERK1/2, and p38[20,21]. To understand the molecular basis of the speculation that irisin stimulates ERK signaling but simultaneously inhibits Smad2/3 phosphorylation, we examined the actions of irisin on TGF-β1 in R1B or R26D cells, which bear a mutation in TGFBR1 or TGFBR2,

respectively[22]. Treatment of R1B cells (lacking functional TGFBR1) with irisin or TGF-β1 produced ERK1/2 phosphorylation but did not increase the phosphorylation of Smad2/3. Notably, the combination of TGF-β1 plus irisin enhanced ERK1/2 phosphorylation (Fig. 6d, middle panel). In contrast, neither TGF-β1 nor irisin phosphorylated ERK1/2 or Smad2/3 in R26G cells, which contain a nonfunctional TGFBR2 (Fig. 6d, right panel). These results indicate that irisin stimulates ERK phosphorylation by functionally activating TGFBR2.

Using confocal microscopy, we detected that FITC-labeled irisin co-localized with TGFBR2 in primary cultures of tubule cells, thereby indicating interaction between irisin and TGFBR2 (Fig. 6e). To test whether irisin interacts with TGFBR2 to inhibit TGF-β1-induced smad2/3 phosphorylation competitively, we examined the dose-response relationships of TGF-β1 and irisin in the primary cultures of tubule cells. As shown in Fig. 6f, TGF-β1 and irisin synergistically stimulated ERK1/2 phosphorylation,

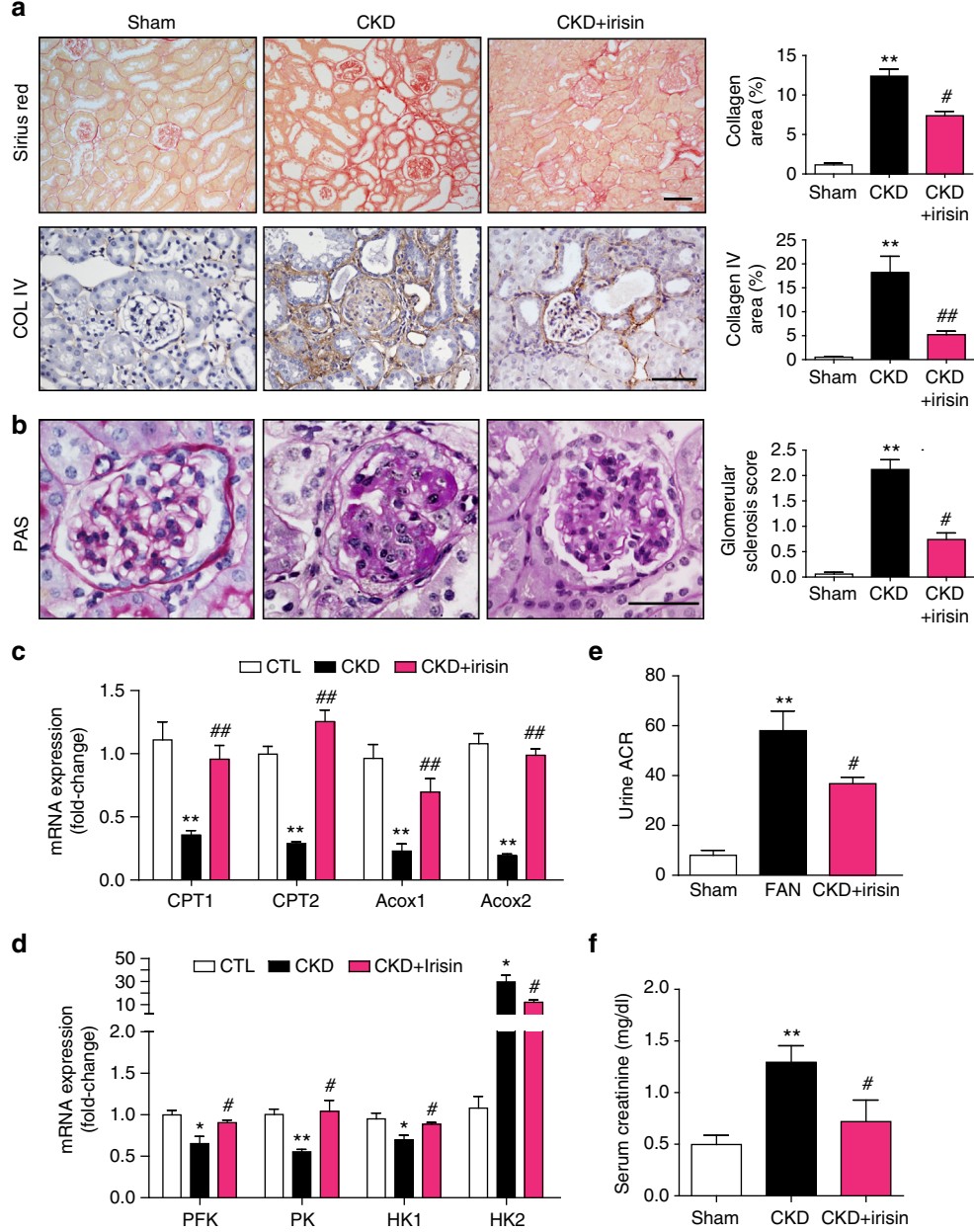

**Fig. 8** Recombinant irisin administration improves kidney function and reduces the development of CKD. **a** Representative images of sirius red staining (upper panel), immunohistochemistry staining for COL IV (lower panel), in subtotal nephrectomy (CKD) mice treated with irisin for 8 weeks. Quantitative analysis of collagen or COL IV in different groups as indicated. $^{**}P < 0.01$ for CKD vs. Sham, $^{\#}P < 0.05$ or $^{\#\#}P < 0.01$ for CKD + irisin vs. CKD ($n = 5$ per group, one-way ANOVA with Bonferroni's multiple comparison test). Scale bars, 50 μm. **b** Representative images of PAS staining in subtotal nephrectomy (CKD) mice treated with or without irisin. Sclerosis index score was shown in left panel. $^{**}P < 0.01$ for CKD vs. Sham, $^{\#}P < 0.05$ for CKD + irisin vs. CKD ($n = 5$ per group, one-way ANOVA with Bonferroni's multiple comparison test). Scale bars, 50 μm. **c** and **d** Real-time qPCR analysis of enzymes expression in glycolysis pathway and fatty acid oxidation pathway. Data were presented as mean ± s.e.m.; $^{*}P < 0.05$ or $^{**}P < 0.01$ for CKD vs. Sham and $^{\#}P < 0.05$ or $^{\#\#}P < 0.01$ for CKD + irisin vs. Sham, $n = 5$ per group (one-way ANOVA with Bonferroni's multiple comparison test). **e** and **f** Kidney function (assessed by serum creatinine and urinary albumin creatinine ratio) was assessed in CKD mice after irisin treatment. Data were presented as mean ± s.e.m. ($n = 5$ per group, $^{**}P < 0.01$ for CKD vs. Sham. $^{\#}$ $P < 0.05$ for CKD + irisin vs CKD, one-way ANOVA with Bonferroni's multiple comparison test)

and as the irisin dose was raised, Smad2/3 phosphorylation decreased. Conversely, when TGFBR2 was overexpressed in tubule cells, irisin-induced inhibition of Smad2/3 phosphorylation was sharply curtailed (Fig. 6g). Because the combination of TGFBR2 and TGFBR1 is required for Smad 2/3 phosphorylation, we concluded that irisin interacts with TGFBR2 to interfere with the ability of TGFBR2 to recruit TGFBR1. The outcome was suppression of Smad2/3 signaling (Fig. 6h).

**Recombinant irisin improves kidney function and blocks fibrosis**. To test whether recombinant irisin protects against the development of renal fibrosis in vivo, we treated mice with folic acid and then administrated irisin (200 μg kg$^{-1}$ day$^{-1}$, ip) for 4 weeks. First, we found that irisin administration resulted in a two-fold increase in serum irisin levels (Fig. 7a). When kidneys were treated with folic acid only, we identified tubular dilatation and atrophy, in addition to fibrosis, in the kidney tubule. In

contrast, in mice treated with irisin and folic acid, renal histopathology and fibrosis improved (Fig. 7b, c), as did kidney function. Improved kidney function was signified by lower serum creatinine levels (Fig. 7d). In addition, these mice developed suppressed expression of fibrotic proteins (fibronectin, collagen 1A, and α-SMA) (Fig. 7e). Along with the amelioration of renal histology and function, mRNA expression of key enzymes in energy metabolism was also improved in damaged kidneys from irisin-treated mice (Fig. 7f, g). To examine whether these responses are limited to folic acid, we studied mice after 4 weeks of subtotal nephrectomy (CKD) by treating them with irisin (200 μg kg$^{-1}$, ip, three times per week) for 8 weeks. As shown in Fig. 8a, kidneys of mice with CKD developed tubular dilatation plus interstitial fibrosis and glomerulosclerosis. Irisin treatment improved both tubular damage and the degree of interstitial fibrosis. On the basis of semi-quantitative sclerosis scoring[23], we found that the degree of glomerular sclerosis was significantly decreased in irisin-treated CKD mice (Fig. 8b). These positive responses were associated with mRNA improvements in the components of energy metabolism (Fig. 8c, d). As expected, irisin treatment also significantly improved albuminuria and serum creatinine as a measure of renal function in mice with CKD (Fig. 8e, f). Thus, irisin administration can improve kidney function in mouse models of CKD or folic acid via improvements in metabolic reprogramming and suppression of fibrosis.

## Discussion

Our major goal is to determine whether muscle-kidney crosstalk can suppress progressive kidney cell damage and subsequent fibrosis. We studied three models of kidney damage (CKD, folic acid administration, and creation of UUO) in mice with muscle-specific PGC-1α overexpression and compared the results with those of littermate, wild type mice receiving the same treatment. In the tubule cells of wild type mice, impaired energy metabolism and tubule damage persisted. These observations are consistent with reports that ATP production is reduced by metabolic reprogramming, which accentuates kidney cell damage and lead to fibrosis in the kidneys. We demonstrated that muscle-specific PGC-1α overexpression corrects metabolic reprogramming, increases ATP production in damaged kidney cells, and suppresses progressive kidney damage and fibrosis. We studied a mPGC-1α transgenic mouse model for two reasons: First, exercise upregulates PGC-1α expression in muscles, mimicking exercise-induced metabolic changes;[3,24] the model might prove the positive benefits of exercise in patients with kidney damage. Second, the model avoids problems assessed in mice because mice with injured kidneys expend different degrees of exercise. Using this model, we uncovered the mediator of muscle-kidney crosstalk: the myokine irisin, which stimulates aerobic metabolism and limits metabolic reprogramming and the extent of kidney cell damage. Irisin also reverses TGF-β-induced suppression of PPARα and PGC-1α. Thus, this sequence of events overcomes metabolic reprogramming, raises ATP production, and reduces tubule cell damage and TGF-β production.

How could exercise influence the metabolism of another organ? Potential mediators of communications between the skeletal muscles and other organs are myokines, also known as muscle-derived cytokines[25]. For example, Boström et al.[26] reported that mPGC-1α mice can stimulate muscle to produce a specific myokine—irisin—which can promote the conversion of white-to-brown adipocytes to stimulate the expression of mitochondrial uncoupling protein-1 (UCP1). Agudelo et al.[6] provided another example of organ-to-organ communication when they uncovered evidence of crosstalk between the skeletal muscles and the brain. Specifically, they found that activation of PGC-1α in

muscles stimulates the expression of kynurenine aminotransferases, reducing the circulating level of kynurenine. Their results indicate that these responses will protect the brain from chronic mild stress. Such results provide evidence that PGC-1α activation in muscles impacts the metabolism of other organs. To the best of our knowledge, there have been no previous reports of crosstalk between skeletal muscles and the kidneys that communicate changes in the metabolism of injured kidney cells to improve their function.

Recent studies have highlighted that tubular cell alter metabolic phenotypes in response to CKD. This metabolic reprogramming is characterized by defects in both glycolysis and fatty acid oxidative pathways. These changes reduce ATP products and enhance stress-induced apoptosis in kidney tubule cells[8,9,27]. To identify the mechanisms by which muscle affects kidney cell bioenergetic functions, we studied mice at 7 days after folic acid administration. This time was chosen because Kang et al.[8] showed that in response to acute kidney damage, after 7 days, cell damage persisted but without significant fibrogenesis in mouse kidneys. For the first time, we show that kidney damage from folic acid affects cellular respiration (noted as extracellular flux) in freshly isolated kidney tubules. In tubules of wild type mice, folic acid damage markedly impaired aerobic respiration. In addition, kidney injury suppressed both ATP-coupled respiration and the spare respiration capacity in tubules from wild type mice.

These results are consistent with recent reports that damaged kidney cells undergo metabolic reprogramming. Strikingly, impairments in the respiration of damaged tubules of wild type mice were significantly corrected in mPGC-1α mice, as evidenced by increased ATP-coupled respiration and spare respiratory capacity in damaged tubules from mPGC-1α mice. Because ATP-coupled respiration and spare respiratory capacity are correlated with improved cell survival[28,29], our results provide evidence that muscle-kidney crosstalk mediated by irisin will prevent progressive damage in kidney tubule cells. Using a metabolomics profiling approach, we also detected a broad range of intermediary metabolites arising from glucose and fatty acid metabolism that changed in response to renal injury. These abnormalities were largely prevented in the injured kidneys of mPGC-1α mice. For example, aerobic glucose metabolism was preserved in injured kidney cells from mPGC-1α mice. This response resulted in an increase in cellular pyruvate, a critical determinant of cell death following kidney injury[30].

With the addition of molecules of serum Fraction 3 (between 10 < and 50 kDa) to primary cultures of kidney cells, cellular energy metabolism improved. This serum fraction from mPGC-1α mice was found to contain the myokine irisin, which was demonstrated to protect kidney cells from TGF-β1-induced metabolic reprogramming and ultimately kidney damage. In fact, an anti-irisin neutralizing antibody blocked the beneficial responses occurring in kidney cells. We recognize that high doses of recombinant irisin did not elicit the same degree of improvement that was achieved by serum from mPGC-1α mice. Presumably, this result may reflect the presence of other myokines or serum factors that improve kidney cell metabolism.

TGF-β1 can suppress PGC-1α, and the latter is a critical transcription factor that regulates mitochondrial biogenesis and the expression of almost all rate-limiting enzymes in the fatty acid oxidation (FAO) pathway. Indeed, we observed that irisin treatment is associated with up-regulated CPT1 and other enzymes in FAO, and this can be explained in the antagonism between irisin and TGF-β1 on smad2/3 signaling. However, we also observed that irisin simulates the expression of PK1 and other key enzymes in the glycolysis pathway, and this response apparently is not involved in Smad3 signaling. It is possible that irisin directly stimulates glycolysis-related gene expression; however, further investigation of the mechanism is required.

We found that irisin mediation of cellular responses can activate ERK and p38. This finding is similar to that of Wang et al., who reported that irisin activates ERK and p38 in adipocytes[31]. The ERK and p38 response can be attributed to the actions of type 2 TGFβ-1 receptors (TGFBR2) because mutation of the TGFBR2 receptor blocks irisin-stimulated phosphorylation of ERK or p38 (Fig. 6). How does irisin influence Smad2/3 phosphorylation? In kidney cells exposed to TGF-β1, addition of irisin suppressed Smad2/3 phosphorylation but enhanced ERK phosphorylation. This leads us to speculate that the interaction between irisin and TGFBR2 interrupts its recruitment of the TGFBR1 receptor. One possibility is that irisin binding to the TGFBR2 receptor stimulates recruitment of another TGFBR2 receptor but prevents TGFBR1 from binding to TGFBR2. This could reflect the fact that TGFBR2 forms homodimers (TGFBR2-TGFBR2) that activate ERK/p38 even when TGFBR1 is absent[32]. Alternatively, irisin could bind to TGFBR2, which stimulates recruitment of unidentified membrane receptors other than TGFBR1. For example, TGFBR2 reportedly recruits the parathyroid hormone type I receptor (PTH1R), and it influences bone formation and hemostasis[33]. In both cases, irisin inhibits TGFBR1 with increased Smad2/3 signaling while continuing to activate ERK and p38 signaling.

In summary, activation of PGC-1α in skeletal muscles increases the production of myokines, including irisin. In three models of kidney cell damage, we found that irisin mediates crosstalk between muscle and kidney tubule cells. Irisin also counteracts metabolic reprogramming in injured kidney cells with improvement in kidney function and suppression of kidney fibrosis. In vivo, recombinant irisin achieved these beneficial effects in damaged kidney cells of wild type mice. Our results could stimulate research into improving communication between the kidneys and other organs while improving the outcome of progressive kidney damage.

## Methods

**Mouse model and sample collection**. Muscle-specific PGC-1α transgenic C57BL/6 mice were purchased from Jackson Laboratories and crossed with CD1 strain breeders. After backcrossing with CD1 mouse for three generations, overexpression of muscle-specific PGC-1α transgene was confirmed (Supplementary Fig. 5a). There was no mPGC-1α leakiness, because the protein level of PGC-1α in kidneys from mPGC-1α mice was identical to values in kidneys from wild type (Supplementary Fig. 5b). The mPGC-1α strain was converted to a CD1 background because CD1 mice usually develop more reliable indices of kidney injury[34,35].

For folic acid nephropathy (FAN), we used 12-week-old male, mPGC-1α mice, the male, littermate wild type mice as controls. 24 mPGC-1α mice or wild type controls were injected intraperitoneally with folic acid (250 mg kg$^{-1}$ in a 0.3 mol L$^{-1}$ NaHCO$_3$ solution). In total 15 mPGC-1α and 15 wild type mice in the sham groups were injected with the same volume of the NaHCO$_3$ solution. There were three wild type and two mPGC-1α mice dead with 5 day after folic acid injection. Same numbers of mice were also used to create the UUO and CKD model[36,37]. For irisin treatment, the possibility that recombinant irisin was contaminated with endotoxin was examined in kidney tubular cells by measuring the phosphorylation of p38in response to irisin or heat-inactivated irisin. Briefly, 20 μl of irisin (200 μg ml$^{-1}$) was heated at 95 °C for 10 min. After centrifugation, the supernatant was add to primary cultures of kidney tubular cells and incubated them for 2 h at 37 °C for 2 h. unheated irisin was examined as a positive control. Cells treated with saline served as negative control. As shown in Supplementary Fig. 6, the unheated irisin stimulated p38 phosphorylation, this response was eliminated in cells treated with heat-inactivated irisin, indicating that endotoxin is very low in the recombinant irisin. We used this unheated recombinant irisin in both in vitro and in vivo experiments. Folic acid-injured mice were injected intraperitoneally with recombinant irisin (200 μg kg$^{-1}$ d$^{-1}$, Phoenix Pharmaceuticals, Inc. Burlingame, CA) for 4 weeks. In mice with CKD, irisin was given for 8 weeks beginning at 4 weeks after subtotal nephrectomy. Urine samples were collected in metabolic cages over 24 h. Blood was collected from anaesthetized mice before mice were perfused via the left ventricle with cooled 0.9% saline. Kidneys and tibialis anterior muscles were collected. All animal experiments were conducted in accordance with Institutional and NIH guidelines using procedures approved by the Baylor College of Medicine Institutional Animal Care and Use Committee.

**Histology examination**. Kidney tissues were fixed in 10% formalin for 48 h at room temperature. Fixed tissues were embedded in paraffin and sectioned into 4-μm-thick slices for Periodic acid-Schiff (PAS) staining and Sirius red staining. For PAS staining (Sigma-Aldrich, MO, USA), after being deparaffinized and hydrated in water, sections were then oxidized for 30 min with 1% periodic acid, washed three times with deionized water, and stained with Schiff's reagent for 30 min, and then washed three times with PBS for 2–3 min. For Sirius red staining, sections were immersed in celestite blue solution for 10 min, followed by being deparaffinized and hydrated into water; after being washed with tap water, sections were immersed in a saturated picronitric acid solution for 30 min and stained by Harris hematoxylin for 30 s. For quantitation of fibrotic area, the images of picrosirius-red-stained sections were captured using Nikon E80 microscope, and quantitative evaluation was performed using NIS-Elements Br 3.0 software (Nikon, Melville, NY). The collagen-stained area was calculated as a percentage of the total area, at least three random-picked kidney cortex images (×200) were examined for each animal.

**Proximal tubules isolation and primary cell cultures**. Mice aged 12 weeks were euthanized by chloral hydrate, and kidneys were collected in a septic environment. Kidney cortices from 12 weeks CD1 mice were dissected visually, minced in small pieces, and digested in 5 mL 37 °C preheated DMEM medium containing 2 mg ml$^{-1}$ type I collagenase and 1% HEPES (Lonza, Walkersville, MD, USA), then placed into a shaking incubator at 37 °C and 5% CO$_2$ for 30 min. Then the filtered tissue suspension was collected by a 100 μm strainer (FALCON®Corning, Durham, USA). The samples were centrifuged (500 rpm, 2 min) to pellet the tubules, washed with 10% FBS-containing DMEM (Mediatech Inc., Manassas, VA, USA), and then centrifuged again. The final pellet, consisting mostly of renal tubules, was resuspended in DMEM medium supplemented with FBS 10%, penicillin 100 IU ml$^{-1}$, and streptomycin 100 μg ml$^{-1}$. The resuspended cells were used to test the cell energy metabolism by Seahorse Bioscience XFe24 cell energy metabolism analyzer (Seahorse Bioscience) directly or incubated at 37 °C in a 5% CO$_2$ incubator with medium changes every 2 d until ~90% confluent. For mitochondrial morphology examination, quiescent primary cultured tubule cells adhering to collagen I-coated coverslips were incubated with MitoSox staining solution (5 μM MitoSox in DMEM medium contains 0.2% FBS) at 37 °C in a 5% CO$_2$ incubator for 30 min, then washed by HBSS solution. The Deconvolution fluorescence microscope was used to photograph the morphology of mitochondria (the integrated microscopy core at the Baylor College of Medicine). To identify irisin colocalizing with TGFBR2 on cell membrane, irisin (1 mg ml$^{-1}$) was labeled with Alexa Fluor 488 Protein Labeling Kit following manufactory instruction (Invitrogen, CA) and then incubated with primary tubule cell cultures for 15 min at 37 °C. After washing three times with PBS, cells were briefly fixed with 2% paraformaldehyde and incubated with mouse anti-TGFBR2 antibody for 2 h at room temperature. After washing, cells were labeled with Alexa 568 anti-mouse IgG secondary antibody and examined with Nikon A1 confocal microscope (Nikon, Melville, NY).

**Kidney fibrotic gene profile**. Total RNA was extracted from kidney tissues with QIAzol Lysis Reagent (QIAGEN, Valencia, CA, USA) and precipitated in isopropanol. complementary DNA (cDNA) was synthesized using an RT2 First Strand kit (QIAGEN Sciences, Maryland, USA) following the manufacturer's instructions and added into an RT2 Profiler™ PCR Array Mouse Fibrosis 96-well plate (QIAGEN Sciences), amplified using a SYBR green superMix for iQ (Quanta Biosciences, Gaithersburg, MD, USA). SYBR Green PCR was performed with the CFX96 System (Bio-Rad), according to the RT2 Profiler PCR Array Handbook. The data were analyzed by PCR Array online analysis software (www.SABiosciences.com) to select the kidney fibrotic gene profile, DAVID Bioinformatics online software was used to do GO analysis (https://david.ncifcrf.gov/), and the KEGG pathway database was used to do pathway analysis (http://www.genome.jp/kegg/pathway.html).

**Serum fraction**. The serum fraction procedure was carried out at 4 °C, unless otherwise indicated. To separate protein fractions by molecular weight, we used various pore sizes of cellulose membrane spin filters (Amicon). Before load samples, all membranes were equilibrated with serum-free medium. Debris in serum was discarded by spinning at 200 g for 15 min. The clear supernatant was loaded onto 100-kDa filter and spun at 4000 g until ~100 μl supernatant was left. Remaining supernatant on the 100-kDa filter was kept for over 100 kDa fraction. The filtrate was loaded onto a 50-kDa filter and spun at 4000×g for 10 min. We added half an input volume of medium and spun it again. The remaining supernatant was adjusted to an input volume with medium and kept for proteins between 50 and 100 kDa fractions. The filtrate was loaded onto a 10-kDa filter and spun at 4000 rpm. The remaining supernatant and filtrate were kept for 10–50 kDa and below10 kDa fractions, respectively. Every sample was aliquoted for mass spectrometry analysis before adjusting the volume with additional medium, and volume-adjusted samples were used for Seahorse assays. Equal volumes of Serum Fraction 3 (10 kd < molecular weight < 50 kD) from mPGC-1α mice were incubated with either mock IgG (10 μg ml$^{-1}$) or the anti-irisin (10 μg ml$^{-1}$) for 4 h at 4 °C. Subsequently, agarose-protein A/G (Santa Crus Biotech) beads were added and incubated for 1 h to remove all IgG from both incubations. The resulting

supernatants were then used to treat primary kidney tubule cells for 12 h before measuring cell respiration.

**Identification of irisin in serum fraction using mass spectra.** The following procedures were performed by the Mass Spectrometry Proteomics Core at the Baylor College of Medicine: SF3 was digested with Trypsin. After vacuum drying, dried peptides were dissolved with a loading solution (5% MeOH, 0.1% FA in water), and irisin peptides were detected by the PRM method using a Fusion™ Tribrid™ mass spectrometer (Thermo Scientifc™). We calculated possible tryptic peptides after FNDC5 fragmentation and filtered in 5–25 amino acid sequences. Among these 11 peptides, 5 peptides belonged to irisin. Half of the digested peptides from each band were analyzed at each machine run with a 4–24% acetonitrile gradient for 45 min. Preselected precursor ions were scanned with 5 min predicted elution windows with 120,000 resolution and 2.0e5 of AGC target by Orbitrap, and then isolated by Quadrupole followed by CID/MS2 analysis. Product ions (MS2) were scanned at 350–1400 m/z with 1.0e4 of the AGC target in rapid mode by Ion Trap. For relative quantification, the raw spectrum file was crunched to an.mgf format by PD1.4, and then imported to Skyline with the raw data file.

**Real- time energy metabolism assessed in primary cultures of mouse renal tubules.** A Seahorse Bioscience XFe24 cell culture microplate (Seahorse Bioscience, North Billerica, USA) was coated with Cell-Tak cell and tissue adhesive (Corning, USA) before 50 μl of 22.4 μg ml$^{-1}$ Cell-Tak solution was added into the well. After 20 min of room temperature incubation, the microplate was washed twice with sterile water. Freshly isolated kidney tubules were resuspended in 37 °C preheated Seahorse XF Assay Medium (Seahorse Bioscience) and seeded onto the coated Seahorse Bioscience XFe24 cell culture microplate at a density of 500 tubules/50 μL/well. A control well had 50 μl Seahorse XF Assay Medium only. Before incubating at 37 °C without CO$_2$ for 25–30 min, the microplate was centrifuged at 200×g for 1 min. A 130-μl preheated analysis buffer was added to the well after ensuring that all tubules sank to the plate's bottom; the incubation was continued at 37 °C without CO$_2$ for 15–20 min. The microplate was then put into a Seahorse XFe24 energy analyzer. An XF Cell Mito Stress Test kit was used to test for mitochondrial stress, an XF Glycolysis Stress Test kit was used to test for glycolysis stress and an XF Palmitate-BSA FAO Substrate was used to test the metabolism of fatty acid oxidation.

**Metabolome analysis of mouse kidney.** A sample homogenization buffer solution was prepared by adding equivalent-volume deionized water into methyl alcohol and putting the buffer solution on ice. A grinding rod and sleeve were washed using methyl alcohol three times, followed deionized water washing three times. A 30–50 mg kidney sample and 7.5 μl mg$^{-1}$ homogenization buffer was added into the washed grinding sleeve. We ground the sample at 300 rpm mp/min for 1 min on ice; the homogenized sample was transferred into a 1.5-ml EP tube and stored at −80 °C. In total 300 μl of methanol was added to 100 μl of the resulting mixture, followed by centrifugation at 15,000×g for 20 min. The resulting supernatants were transferred into autosampler vials and 5.0 μl was injected into a 6490-triple quadrupole mass spectrometer for analysis (Agilent Technologies, Santa Clara, CA, USA) coupled to an HPLC system (Agilent Technologies) via the multiple reaction monitoring method. The drying gas temperature was set up at 250 °C for the positive mode and 290 °C for the negative mode. Drying and sheath gas flow were maintained at 14 l/min and 12 l/min for both modes, respectively. Capillary voltage was set at 3500 V for the positive mode and 3200 V for the negative mode. Fatty acid separation was achieved using a 1260 Infinity Binary LC system equipped with a 150 mm × 2.0 mm (Luna 3u Phenyl-Hexyl, Phenomenex) column. The column temperature was maintained at 40 °C, and the flow rate was 0.2 ml min$^{-1}$ with a gradient in a 37-min run. Gradients were run starting from 60% buffer A containing ammonia acetate (water/methanol, 80:20 v/v, pH = 8.4) and 40% buffer B containing ammonia acetate (acetonitrile/water 90:10 v/v, pH = 8.4) and to 50% A for 0–8 min; 50% A to 33% A for 8–13 min; 33% A was held for 13–22 min; 33% A to 0% A for 22–23 min; 100% B was held for 23–29 min; 0% A to 60% A for 29–30 min; 60% A was held for 7 min to re-equilibrate the column. The fatty acids were measured by MS in negative mode with electrospray ionization.

Carnitines were separated using an LC system equipped with a 100 × 2.1 mm$^2$ (Agilent XDB C18) column. The column temperature was maintained at 4 °C, and the flow rate was 0.2 ml min$^{-1}$ with a gradient in a 25-min run. Gradients were run starting from 98% buffer A (water, 0.1% formic acid) and 2% buffer B (acetonitrile, 0.1% formic acid) and to 80% A for 0–6 min; 80% A to 2% A for 6–15 min; 2% A was held for 15–18 min; 2% A to 98% A for 18–20 min; 98% A was held for 5 min to re-equilibrate the column. Carnitines were measured by MS in positive mode with electrospray ionization. The peak area for each metabolite was integrated using MassHunter Workstation Software Quantitative Analysis Version B.06.00 (Agilent Technologies). All procedures were carried out by Metabolomics Core at the Baylor College of Medicine.

**Measurement of ATP in tissues/cells.** ATP in tissues/cells was measured using an ATP colorimetric/fluorometric assay kit (Biovision Inc., Milpitas, USA) on 1 × 10$^6$ cells or homogenized tissues (10 mg) in 100 μl ATP assay buffer. A

deproteinization sample preparation kit (Biovision) was used to deproteinize cell lysate or tissue homogenate according to instructions. We added 2–50 μl of deproteinzed samples to a 96-well plate and adjusted the volume to 50 μl per well with ATP assay buffer, and a standard curve was prepared following the kit's instructions. Then a 50-μl reaction mixture that contains a 44-μl ATP assay buffer, 2-μl ATP probe, 2-μl ATP converter, and 2-μl developer was added to each well containing the ATP standard and test samples. The plate was incubated at room temperature for 30 min after mixing well, and protected from light. A FLUOstar Omega microplate reader (BMG Labtech, USA) was used to measure the absorbance at 570 nm. A correct background was determined by subtracting the value derived from the 0 ATP standard from all readings. We applied ATP sample readings to the standard curve to calculate the concentration of ATP in the sample well using the formula: sample ATP concentration (c) = B (ATP amount in the reaction well from standard curve)/V (the sample volume added into sample wells) × D (the dilution factor); ATP amount in the sample well (B) = ATP Std × (ODsample)/(ODsample + ATPstd− ODsample) (pmol).

**Serum creatinine measurement.** Blood samples were taken from the WT and mPGC-1α mice. After clotting at room temperature for 1 h, blood was centrifuged at 3000 rpm for 10 min to obtain serum. A creatinine (serum) colorimetric assay kit (Cayman Biochemical) was used to measure serum creatinine. A standard curve was made using the creatinine standard according to the kit's instructions. Some 15-μl serum samples were added into a 96-well solid plate; a 100-μl creatinine sodium borate, creatinine surfactant, and creatinine sodium hydroxide (2:6:4) mixture was added subsequently, and then 100 μl creatinine color reagent was added. A FLUOstar Omega microplate reader (BMG Labtech) read the initial absorbance at 495 nm when there was a reaction for 1 min and reread the final absorbance at 495 nm, when there was a reaction for 7 min. The creatinine concentration was calculated based on the standard curve.

**Urinary albumin/creatinine measurement.** A creatinine (urinary) Colorimetric Assay Kit (Cayman Biochemical) was used to measure urine creatinine. A standard curve was made using the creatinine standard according to the kit's instructions. Urine samples were diluted 10-fold, and 15-μl samples were added to a 96-well plate, and 150 μl creatinine sodium borate, creatinine surfactant, creatinine color reagent, and creatinine sodium hydroxide (2:6:10:3.6) mixture was added subsequently. After shaking the plate for 10 min at room temperature, a FLUOstar Omega microplate reader (BMG Labtech) read the initial absorbance at 495 nm, and then added 5 μl creatinine acid solution into every well to stop reactions. We then shook the plate for 20 min at room temperature and read the final absorbance at 495 nm. The creatinine concentration was calculated based on the standard curve. Urinary albumin was determined using an Albuwell Mouse Urinary Albumin ELISA kit (Exocell Inc., Philadelphia, PA, USA) according to the manufacturer's protocol. Proteinuria was expressed as the urinary albumin/creatinine ratio.

**Western blots.** Total proteins from kidneys were extracted by RIPA lysis and extraction buffer (G-Biosciences, MO, USA) containing protease and phosphatase inhibitor (Thermo Scientific, Rockford, IL, USA) on ice. The lysates were collected after centrifuging at 13,600 rpm at 4 °C for 15 min and heated with a sample buffer at 100 °C for 5 min before being separated by sodium dodecyl sulfate–polyacrylamide gel electrophoresis (SDS–PAGE) on gradient gels. The proteins were electro transferred to polyvinylidene difluoride membranes (Merck Millipore Ltd, Darmstadt, Germany). The membranes were blocked with 5% non-fat milk-TBS and incubated overnight with primary antibodies at 4 °C, followed by 1 h of incubation with dylight 680 or 800 conjugated secondary antibodies (Cell Signaling Technology, USA) at room temperature. The bands were visualized with the Odyssey® infrared imaging system (LI-COR Biosciences, Nebraska, USA). To ensure equal protein loading, the β-actin protein was used as the endogenous control.

The anti-collagen I antibody was purchased from Millipore (EMD Millipore, Darmstadt, Germany). The anti-a-SMA antibody was purchased from Sigma-Aldrich. The anti-CTGF, anti-FN and anti-TGFR I/II antibodies were purchased from Santa Cruz Biotechnology (Santa Cruz, CA, USA). Other antibodies were purchased from Cell Signaling Technology. Uncropped western blots are provided in Supplementary Fig. 8.

**mRNA preparation and quantitative real-time RT-PCR.** Quantitative analysis of the target mRNA expression was performed with real-time PCR by the relative standard curve method. Total RNA was extracted from snap-frozen kidney/tibialis anterior muscle tissues with a QIAzol lysis reagent (QIAGEN, Valencia, CA, USA) and precipitated in isopropanol. cDNA was synthesized using qScript cDNA SuperMix and amplified using B-R SYBR green superMix for iQ (Quanta Biosciences). SYBR Green real-time quantitative PCR was performed with the CFX96 System (Bio-Rad), according to the manufacturer's instructions. The specificity of real-time PCR was conformed using melting–curve analysis. The expression levels of the target genes from the muscles were normalized by the glyceraldehyde 3-phosphate dehydrogenase (GAPDH) level or ribosomal protein L39 (rpl39) in each sample. All sequences of primers are listed in Supplementary Fig. 7.

**Statistics**. Data are shown as mean ± s.e.m. and were evaluated using GraphPad Prism 6. For experiments comparing two groups, we analyzed results by the Student–Newman–Kuel's two-tailed, unpaired tests. When more than two groups were compared, ANOVA followed by Bonferroni's multiple comparison test were used to analyze differences between two interested groups. A value of $P < 0.05$ was considered statistically significant.

**Data availability**. Data generated or analyzed during this study are included in this published article and in Supplementary Information files. Data are also available from the corresponding author upon request.

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

## Acknowledgements

This work was supported by National Institutes of Health grants (5RO1-AR063686 to Z.H.). H.P. is supported by the National Natural Science Foundation of China (NSFC 81670675, NSFC 81470955 to H.P.) and China Scholarship Council. Q.Q.W. is supported by International Program for Ph.D. Candidates, Sun Yat-Sen University. This research was supported by the funds from Metabolomic Core Facility, Alkek Center for Molecular Discovery, Baylor College of Medicine. We thank Dr. J.M. (Memorial Sloan-Kettering Cancer Center) for the generous gift of DR26 and R1B cell lines. These experiments were partially supported by Dr. and Mrs. H.S., the founder of Selzman Institute for Kidney Health at BCM.

## Author contributions

H.P., Q.W., Y.W., Z.H. carried out experiments, study design and data analysis. J.Q. and S.y.J. performed experiments relating to mass spectra and data analysis. F.L. and V.S. performed metabolomics experiments and results analysis. H.P., T.L., J.Q., X.-h.F., W.E.M., B.H.G. and Z.H. interpreted the data, and H.P. and Z.H. wrote the manuscript.

## Additional information

**Competing interests:** The authors declare no competing financial interests.

