## [Peer Review file · Nature Communications]

Reviewers' comments:

Reviewer #1 (Remarks to the Author):

This paper reports on the fact that mice with elevated PGC1 α in muscle have resistance to kidney disease in mice. The muscle-specific transgenic mice have characteristics of chronic exercise so this is a potentially interesting finding. Overall, the authors have a rather interesting result but my enthusiasm would go higher if this were done more quantitatively in several important aspects as detailed below.

1. Overall, the paper does not read well. There are many grammatical mistakes so this needs work from someone more fluent in English.
2. Throughout the paper, have the statistics been corrected for multiple hypothesis testing? It is not clear.
3. The relationship between metabolic programming and the kidney disease and its improvement are not clear. Where has this data gone beyond the expected increases in oxidative metabolism expected in the PGC1 α transgenic mice.
4. The effects of the anti-irisin antibody should be accompanied by a suitable control antibody. IgG themselves can have effects on many processes so a vehicle control alone is not enough.
5. The key findings with recombinant irisin should contain a dose response at least in cultured cell experiments. Also, for in vivo work, it would be very helpful to know how the protein was made and what were the levels of purity and endotoxin. This is especially an issue with proteins made in bacteria, obviously.

Reviewer #2 (Remarks to the Author):

The study by Peng et al "Myokines mediate muscle-kidney crosstalk suppressing metabolic reprogramming and interstitial fibrosis" describes a novel important metabolic cross-talk between the muscle and the kidney.

They show that mice carrying transgenic over-expression of PGC1 α in the muscle confers resistance to metabolic reprogramming normally observed in chronic kidney diseases and fibrosis, i.e. defective FAO. Notably, the study is conducted on three different models of CKD: folic acid treatment (FAN), unilateral ureteral obstruction (UUO) and subtotal nephrectomy (CKD). The authors move on to demonstrate that a circulating factor released by the muscles of these mice is responsible for this protection and they identify among the myokines contained in the fraction which is able to reprogram the metabolism of renal epithelial cells in seahorse experiments that irisin is one such key factor. Although irisin has been questioned as a possible myokine involved in improving metabolism of a number of organs, the authors convincingly show the central role of this myokine in the protection to chronic kidney disease. They identify the peptide corresponding to irisin by mass spectrometry, they show that antibodies against irisin injected into PGC1 α - muscle transgenic mice revert their protection from CKD and finally they show that irisin injection in WT mice is sufficient to confer protection to metabolic reprogramming and to CKD. The authors move on to identify the receptor enabling renal epithelial cells to respond to irisin and they show that this is TGF β 2 receptor, but not TGF β 1 receptor.

I would like first of all to congratulate the authors for an impressive piece of work. It is really hard for this reviewer to find experiments missing or data that are not of superb quality. I would only encourage the authors to include more information on the characterization of the mice themselves. Even if they were reported before, it would be essential to control for lack of leakage of expression of the transgene especially in the kidney. The effect is so prominent that it is important to demonstrate that it is secondary to an exclusive expression in the muscle.

Reviewer #3 (Remarks to the Author):

This is a very interesting paper by Peng et al (Sun Yat-sen and Baylor Universities) that strives to develop a story in mouse models of chronic kidney injury that under simulated exercise conditions (induced using mice with muscle-specific PPAR gamma coactivator-1 [PGC-1 alpha] over-expression, myokines (specifically the irisin precursor FNDC5) are up-regulated in the "exercised" muscle, cleaved by an unknown protease and released into the circulation and mediated remarkable renoprotective effects. Data are presented to suggest a mechanistic pathway involving tubular cell metabolic reprogramming, transforming growth factor beta (TGF beta) signaling and kidney fibrosis. The potential clinical implications are intriguing, although various human studies suggest that the elevation of serum levels with acute exercise normalize quickly once the exercise is over and chronic exercise may actually be associated with lower serum irisin levels, so clearly more data are needed from both human and animal studies. Irisin was discovered in 2012 and there are already 456 "irisin" publications in PubMed but none on fibrosis, making this work novel. The field is currently challenged by many unknown – the identity of the FNDC5-cleaving protease, whether there is a specific irisin cellular receptor and the lack of a detailed survey of tissues that are able to produce irisin. One paper made reference to its presence in skin, eye and thyroid in addition to muscle. Overall the study is potentially very interesting but there are some significant concerns with the study design and data analyses that need to be addressed.

SPECIFIC COMMENTS

1. As baseline studies, FNDC5 and irisin levels should be measured in the kidneys and serum in all study groups (including shams).
2. A reference needs to be provided for the mPGC-1alpha transgenic mice. What is the rationale for only back-crossing the original C57BL/6 line onto the CD1 line for only 3 generations before these studies began – this is very short? Were sibling pairs used to minimize the potential for background strain heterogeneity? Was the genotype/irisin-over-producing phenotype verified? Were all studies performed in male mice?
3. A careful review of the reference citations is required – several do not match the text.
4. The energy metabolomics studies in tubules and kidney tissue are interesting but tend to stand alone as descriptive observations that are not clearly linked mechanistically with the anti-fibrotic effects of irisin. The implication is that the observed metabolic changes are associated with less TGF-beta production but this possibility is not adequately pursued.
5. Overall there is a lack of quantitative data analytics. The sample size provided in the Figure Legends suggests a range; many immunoblots show single samples for each of the various experimental conditions. Numbers should be clearly stated for each experiment.
6. Figure 1
 - a. What was the mortality rate in the folic acid-injected mice?
 - b. The degree of fibrosis seen histologically should be evaluated quantitatively.
 - c. Gene labels in Figure 1C are difficult to read.
 - d. Were there any differences in gene expression profiles between the two sham groups?
7. Figure 2
 - a. The legend suggests 5 mice/group; there are more than 5 points for each of the groups in Figure 2A.
8. Figure 3
 - a. A comment is on the differences between the sham groups in Figure 3D.
 - b. For Figures 3E-J it is inferred that results using whole kidney samples reflect tubular metabolic profiles. Has this assumption been validated? Were these all based in 14 day folic acid nephropathy experiments?
9. Figure 4
 - a. Figure 4D represents muscle mRNA levels. Were the myokine mRNA levels that were significantly elevated in the mPGC-1 alpha mice all shown to be elevated in the serum?
10. Figure 5
 - a. What was the rationale for the culture conditions shown? Were other irisin doses and incubation times tested?

11. Figure 6

- a. These are a key set of mechanistic studies that look like they were all n=1 for each specific experimental condition. They key observations need to be performed with adequate numbers to perform reliable quantitative analyses.
- b. Can irisin be co-immunoprecipitated with one or both of the TGF-beta receptors?
- c. Can irisin be detected immunohistochemically in the kidney tissue and can it be co-localized with TGF-beta receptors?
- d. In current working models of kidney fibrosis it is the ability of TGF-beta to activate/transform interstitial fibroblasts and pericytes that drives matrix production and fibrosis. Relevant key experiments should be repeated using fibroblasts and include assessment of alpha SMA and matrix proteins in addition to ERK and Smad phosphorylation.

12. Figure 7

- a. Was proteinuria reduced as was reported in the experiments in Figure 2SE?
- b. Can irisin be detected in the urine?

Reviewers' comments:

Reviewer #1 (Remarks to the Author):

This paper reports on the fact that mice with elevated PGC1 α in muscle have resistance to kidney disease in mice. The muscle-specific transgenic mice have characteristics of chronic exercise so this is a potentially interesting finding. Overall, the authors have a rather interesting result but my enthusiasm would go higher if this were done more quantitatively in several important aspects as detailed below.

1. Overall, the paper does not read well. There are many grammatical mistakes so this needs work from someone more fluent in English.

Answer: we have reviewed our manuscript and corrected grammatical errors.

2. Throughout the paper, have the statistics been corrected for multiple hypothesis testing?

Answer: as the Reviewer requests, we used ANOVA with Bonferroni correction for comparisons among more than two groups (see Methods, Statistics, page 25)

3. The relationship between metabolic programming and the kidney disease and its improvement are not clear. Where has this data gone beyond the expected increases in oxidative metabolism expected in the PGC1 α transgenic mice.

Answer: in our experiments, metabolic programming was followed with more severe kidney cell damage (Figure 1 and 3). The myokine, irisin, suppressed these responses and improved metabolic reprogramming (Figure 5 and Figure 7). Other reports have linked impaired energy metabolism (metabolic reprogramming) to the degree of kidney cell damage. For example, Rowe et al., reported that blocking glycolysis will suppress degeneration of tubule cells from polycystic kidneys (Nat Med 2013). In addition, investigators concluded that kidney injury-induced metabolic reprogramming in kidney cells leads to cell injury and subsequently, renal fibrosis (Kang, H. M., et al Nat Med 2015. Tran MT. et al, Nature 2016). We also found that irisin suppresses the development of kidney fibrosis (Figure 7&8). These results are included in the Discussion (please see text highlighted in red, page 17).

4. The effects of the anti-irisin antibody should be accompanied by a suitable control antibody. IgG themselves can have effects on many processes so a vehicle control alone is not enough.

Answer: In Method on page 23, we have detailed information about the procedures based on the anti-irisin antibody. In short, equal volumes of Serum fraction 3 from mPGC-1 α mice were incubated with either mock IgG or the anti-irisin. Subsequently, agarose-protein A/G beads were used to remove all IgG from both incubations. The resulting supernatants were then used to treat kidney tubule cells. Serum samples treated with the anti-irisin antibody lost the ability to stimulate oxygen consumption in tubule (Figure 4f).

5. The key findings with recombinant irisin should contain a dose response at least in cultured cell experiments. Also, for in vivo work, it would be very helpful to know how the protein was made and what were the levels of purity and endotoxin. This is especially an issue with proteins made in bacteria, obviously.

Answer: In Figure 6f, we show the dose-response relationship for the recombinant irisin we studied. It was purchased from (Phoenix Pharmaceuticals, Inc. Burlingame, CA). The purity $\geq 95\%$. The manufacturing was based on use of a SUMOpro-Tag system and endotoxin activity was not test in this products.

To address reviewer's question, we heated irisin for 10 min at 95°C, and found the ability of irisin to activate p38 is eliminated. Because endotoxin (LPS) is heat-resistant molecule, we concluded that endotoxin level is very low in the recombinant irisin. Please Figure A below:

A: Heat-inactivated irisin did not stimulate p38 phosphorylation

Primary kidney tubule cell cultures were treated with PBS, irisin (2ug/ml) or irisin but boiled for 10 min before treatment. After incubation at 37°C for 1h, cells were lysis in RIPA buffer and the cell lysate was subjected to western blotting using anti p-p38 antibody, the non-phosphorylated p38 was used as loading control. Because endotoxin is heat resistant, the result indicated that the recombinant irisin has very low level of endotoxin.

Reviewer #2 (Remarks to the Author):

I would like first of all to congratulate the authors for an impressive piece of work. It is really hard for this reviewer to find experiments missing or data that are not of superb quality. I would only encourage the authors to include more information on the characterization of the mice themselves.

Answer: we thank the reviewer for his/her positive comments and encouragement. The muscle-specific PGC-1 α transgenic (mPGC-1 α) mouse was created by the Lin et al., (Nature. 2002 Aug 15;418:797-801). To evaluate these mice for “leakage”, we confirmed overexpression of PGC-1 α in muscles by western blotting but there was no increase in PGC-1 α expression in the kidneys of mPGC-1 α mice compared to littermate, wildtype control mice. These results were also added to Supplementary Figure 5.

Reviewer #3 (Remarks to the Author):

This is a very interesting paper by Peng et al (Sun Yat-sen and Baylor Universities)... Overall the study is potentially very interesting but there are some significant concerns with the study design and data analyses that need to be addressed... ”.

Answer:

We appreciate the reviewer's encouragement and his/her comments regarding this complicated area of investigation.

We apologize for the error we discovered in our Reference Software and have corrected the problem.

Specific Comments, Reviewer 3:

1. As baseline studies, FNDC5 and irisin levels should be measured in the kidneys and serum in all study groups (including shams).

Answer: as suggested, we have added the serum levels of irisin in each group of experimental mice (Figure 4b and Figure 7a). We also have added the mRNA levels of FNDC5 in kidneys and found the levels were very low vs. values found in muscles of wild type. Please see attached Figure B below:

B

B: low expression of irisin in kidney

Kidney tissues from wild type mice or wild type mice treated with folic acid (7days) were subjected to a real-time qPCR to detect the mRNA expression of FNDC5, a precursor of irisin. β-actin was used as a loading control. Compared with its mRNA level in skeletal muscle, the FNDC5 expression is very low in kidneys, indicating irisin is not an abundantly expressed gene in kidney. Data was presented as mean± s.e.m.; **p<0.01 vs. sham control kidney or kidney from mice with FAN, n=3 per group (one way ANOVA).

2. A reference needs to be provided for the mPGC-1alpha transgenic mice. What is the rationale for only back-crossing the original C57BL/6 line onto the CD1 line for only 3 generations before these studies began – this is very short? Were sibling pairs used to minimize the potential for background stain heterogeneity? Was the genotype/irisin-over-producing phenotype verified? Were all studies performed in male mice?

Answer: we have added the reference describing the creation of mPGC-1α mice (Nature. 2002 Aug 15;418:797-801). Regarding the back-crossing procedure, we decided to create mPGC-1α in CD1 background mice because C57BL/6 mice are somewhat resistant to kidney damage. Therefore, we created mPGC-1α mice in a CD1 background to obtain a more reliable model of kidney damage following subtotal nephrectomy or folic acid administration (Fogo A, et al Kidney Int. 64, 2003; Leelahavanichkul, A. et al Kidney Int. 78, 2010; Wibke Bechtel, Nat Med 16, 2010 May). We back-crossed the original C57BL/6 mice bearing the mPGC-1α transgene onto the CD1 line for 3 generations over almost a year of work. We initially confirmed that the transgene was functioning by detecting over-expression of PGC-1α in muscle of the CD1 PGC-1α mice we created. We also documented that serum levels of irisin are increased compared to results from wild type, littermate control mice (Figure 4b). In our experiments, we used male, mPGC-1α mice and plan to study responses in female mice.

3. A careful review of the reference citations is required – several do not match the text.

Answer: we have corrected the problem.

4. The energy metabolomics studies in tubules and kidney tissue are interesting but tend to stand alone as descriptive observations that are not clearly linked mechanistically with the anti-fibrotic effects of irisin. The implication is that the observed metabolic changes are associated with less TGF-beta production but this possibility is not adequately pursued.

Answer: we decided to study 3 models of kidney damage (CKD, folic acid administration and creation of UUO) to determine responses that are occurring in progressive kidney damage. In conditions we studied, impaired energy metabolism persisted, damaging kidney cells. Our observations are consistent with reports that reduced ATP production (metabolic reprogramming) accentuates kidney cell damage and leads to fibrosis in the kidney (Kang, H. M., et al Nat Med 2015. Venkatachalam, M. A., et al J.Am.Soc.Nephrol. 2015. Zuk A. and Bonventre JV. Annu.Rev.Med. 2016). We also have demonstrated that irisin stimulates aerobic metabolism and limits metabolic reprogramming and the extent of cell damage (Figure 4, 5). As discussed (Page 18) our results indicated that kidney injury in wild type mice exhibits an increase in TGF- β and metabolic reprogramming resulting in high grade kidney fibrosis. In contrast, these responses were suppressed in mPGC-1 α mice. To identify the mechanism underlying our observations, we studied primary cultures of kidney tubule cells: irisin treatment reversed TGF- β -induced suppression of PPAR α and PGC-1 α . This sequence of events overcomes metabolic reprogramming, raising ATP production, reducing tubule cell damage and TGF- β production.

5. Overall there is a lack of quantitative data analytics. The sample size provided in the Figure Legends suggests a range; many immunoblots show single samples for each of the various experimental conditions. Numbers should be clearly stated for each experiment.

Answer: for each immunoblot figure based on results from isolated cells, we have added the number of independent experiments to the legend of Figure 6. For results obtained in mice, we have added the number of mice studied in each experiment (Figures 1b, 2a, 7c & 8a).

6. a. What was the mortality rate in the folic acid-injected mice?

Answer: 24 mPGC-1 α , male mice and 24 wild type, littermate male, control mice were received same dose of folic acid, there 2 mice in mPGC-1 α group (8%) and 3 mice in control group (12%) were dead within 7 days after Folic acid injection (Method, page 20).

b. The degree of fibrosis seen histologically should be evaluated quantitatively.

Answer: as described in Methods (page 21), we measured the areas stained by Sirius Red in injured kidneys using NIS-Elements Br 3.0 software (Nikon) in order to quantify the degree of fibrosis (Figures 1b, 7c and 8a).

c. Gene labels in Figure 1C are difficult to read.

Answer: we have improved the resolution of this image (Figure 1c).

d. Were there any differences in gene expression profiles between the two sham groups?

Answer: we have added the analysis of the Sham groups in upper panel of Figure 1d.

7. Figure 2

a. The legend suggests 5 mice/group; there are more the 5 points for each of the groups in Figure 2A.

Answer: in this experiment we sought to determine if there was evidence for damage in kidneys of wild type vs. the mPGC-1 α mice. Four images were obtained for each mouse and there were 5 mice in each group. Thus, for each group of mice, there were 20 points. To avoid confusion, we have calculated average values for each mouse and present the results in a bar graph.

8. Figure 3

a. A comment is on the differences between the sham groups in Figure 3D.

Answer: By definition, the difference between ATP produced by mitochondrial oxidative phosphorylation at basal and that at maximal activity is termed “spare respiratory capacity”. The basal OCR represents the oxidative phosphorylation at basal state and Maxima OCR represents the oxidative phosphorylation at maximal active level. Kidney cells can require a sudden burst of additional cellular energy in response to stress or damage. If spare respiratory capacity of the cells is not sufficient to provide the required ATP affected cells risk being driven into cell death. Exhaustion of the spare respiratory capacity has been correlated with cell viability in a variety of pathologies including heart diseases and neurodegenerative disorders (Sansbury, BE et al. *Chemico-Biological Interactions*, vol. 191, pp. 288–295, 2011. Yadava N and Nicholls DG, *J of Neuroscience*, vol. 27, pp. 7310–7317, 2007). In our study, we found that spare respiratory capacity is increased in kidney tubules and primary tubular cells cultures treated with irisin. This can explain why there are less cell death or damage when mPGC-1 α mice were treated with folic acid. We added this comment into Discussion, see page 17.

b. For Figures 3E-J it is inferred that results using whole kidney samples reflect tubular metabolic profiles. Has this assumption been validated? Were these all based in 14day folic acid nephropathy experiments?

Answer: our goal was to determine if responses in muscle will suppress folic acid-induced kidney injury. We find there are similar metabolic profiles in mouse kidneys and in tubule cells but have not formally compared the magnitude of these measurements. Regarding the protocol energy metabolism, we studied mice at 7 days after folic acid administration. This time was chosen based on the reports of Dr. Susztak’s group who showed there was severe tubular cell damage but no obvious pathologic fibrosis in mouse kidneys at 5 or 7 days after folic acid (Kang, H. M., et al *Nat Med* 2015). To assess the outcome of kidney injury, we examined mice at 14 days after folic acid and found advanced fibrosis (Figure 1). These results led us to study how myokines might benefit mouse kidney cell damage at early stage following folic acid injection. At the 7 day point, there is ongoing damage to tubule cells but minimal fibrosis and there is the possibility that damage to kidney cells could be reversed.

9. Figure 4

a. Figure 4D represents muscle mRNA levels. Were the myokine mRNA levels that were significantly elevated in the mPGC-1 alpha mice all shown to be elevated in the serum?

Answer: we measured mRNA expression of 24 myokines present in muscles of wild type and mPGC-1 α mice (Figure 4a). Using ELISA, we detected serum levels of 3 of these myokines (irisin, IL-15, BdnF) which were elevated in serum of mPGC-1 α mice (Figure 4b).

10. Figure 5

a. What was the rationale for the culture conditions shown? Were other irisin doses and incubation times tested?

Answer: we studied responses of primary cultures of kidney tubule cells treated with TGF-1 β because it is involved in metabolic reprogramming (Kang, H. M., et al Nat Med 2015). We also decided to study the influence of irisin because it is reported that it regulates mitochondrial function of adipocytes (Bostrom, P et al. Nature 481, 463-468 2012). Regarding irisin dosage, we tested different concentrations of irisin in primary tubule cell cultures and found dose-dependent suppression of TGF- β 1 activity (Figure 6f). In experiments with primary cultures of cells, we incubated them with irisin for 1 hour to examine cellular respiration. We also evaluated images of mitochondria following 24 h of incubation with irisin. We added new results of PPAR α and PGC-1 α that are more relevance for this study (Figure 5g).

11. Figure 6

a. These are a key set of mechanistic studies that look like they were all n=1 for each specific experimental condition. They key observations need to be performed with adequate numbers to perform reliable quantitative analyses.

Answer: for the cultured cell results in Figure 6, all experiments were independently performed 3 times and this information is added to the figure legends.

b. Can irisin be co-immunoprecipitated with one or both of the TGF-beta receptors?

Answer: our speculation about irisin interacting with TGFBR2 is based on responses we obtained from mutated TGF- β receptors (Figure 6d). There is technique issue to co-immunoprecipitate TGFBR2 with its ligands. Instead we performed a co-localization analysis as the reviewer suggested (see below).

c. Can irisin be detected immunohistochemically in the kidney tissue and can it be co-localized with TGF-beta receptors?

Answer: We performed immunofluorescence staining and confocal microscopy to detect FITC-labelled irisin which is colocalized with the TGFBR2 in primary cultured tubule cells (Figure 6e).

d. In current working models of kidney fibrosis it is the ability of TGF-beta to activate/transform interstitial fibroblasts and pericytes that drives matrix production and fibrosis. Relevant key experiments should be repeated using fibroblasts and include assessment of alpha SMA and matrix proteins in addition to ERK and Smad phosphorylation.

Answer: as noted, our goal is to determine if responses in muscle can suppress folic acid-induced kidney injury and if this protective response will occur in CKD or UUO. We did find a protective effect of myokines following folic acid damage as well as in CKD or UUO. The protection against progressive kidney cell damage was detected before there was substantial fibrosis (Figure 2 & 3). In short, we have not claimed to reverse fibrosis in the damaged kidneys. Instead, our results show that myokines can protect kidney tubule cells from progressive damage and metabolic reprogramming. Since the energy metabolism of tubular cells and fibroblasts are

quite different, we have not investigated how myokines might suppress metabolic reprogramming in fibroblast. Unfortunately, we do not have sufficient resources to address this new topic. Instead we will incorporate the reviewers' suggestion into our future investigation.

12. Figure 7

a. was proteinuria reduced as was reported in the experiments in Figure 2SE?

Answer: We added the result as the reviewer requested (Figure 8b).

b. Can irisin be detected in the urine?

Answer: we did not examine urinary irisin.

REVIEWERS' COMMENTS:

Reviewer #2 (Remarks to the Author):

The authors have addressed my only concern and further strengthened their study by responding to other comments of the reviewers.

Reviewer #2 has commented also on Reviewer #1 concerns because they could not assist us in this round of revision. Below their comments:

The authors show in Figure A of the response letter to reviewer#1 that indeed the irisin used in their study and provided by Phoenix Pharmaceuticals contains a small, but detectable, amount of endotoxins as evidenced by the capability to induce p38 in primary kidney tubule cells. The authors convincingly deduce that the LPS levels are very low because the activation of p38 is not huge, yet this activation is reduced to baseline when the irisin preparation is heated. The question is whether or not the irisin used in the study was heat-inactivated prior to the studies both in the in vitro and in the in vivo experiments. I could not find any detail in the methods and would suggest that the authors state clearly this point both if they did or if they did not heat-inactivated the preparation. In the last case, I would also state that the irisin preparation did contain some endotoxin contamination, albeit minimal.

Reviewer #3 (Remarks to the Author):

The authors have adequately addressed my concerns.

Response to Reviewer:

The specific comment of Reviewers:

The authors show in Figure A of the response letter to reviewer#1 that indeed the irisin used in their study and provided by Phoenix Pharmaceuticals contains a small, but detectable, amount of endotoxins as evidenced by the capability to induce p38 in primary kidney tubule cells. The authors convincingly deduce that the LPS levels are very low because the activation of p38 is not huge, yet this activation is reduced to baseline when the irisin preparation is heated. The question is whether or not the irisin used in the study was heat-inactivated prior to the studies both in the in vitro and in the in vivo experiments. I could not find any detail in the methods and would suggest that the authors state clearly this point both if they did and if they did not heat-inactivated the preparation. In the last case, I would also state that the irisin preparation did contain some endotoxin contamination, albeit minimal.

Answer: the possibility that recombinant irisin was contaminated with endotoxin was examined in kidney tubular cells by measuring the phosphorylation of p38 in response to irisin or heat-inactivated irisin. This method was used because endotoxin is a heat-resistant molecule.

Briefly, 20 μ l of irisin (200 μ g/ml) was heated at 95°C for 10 min. After centrifugation, the supernatant was added to primary cultures of kidney tubular cells and incubated them for 2 h at 37°C for 2 h. unheated irisin was examined as a positive control. Cells treated with saline served as negative control. As shown in supplemental figure 6, the unheated irisin stimulated p38 phosphorylation, this response was eliminated in cells treated with heat-inactivated irisin. Based on these results, we conclude that endotoxin is very low in the recombinant irisin. Thus, we used this unheated recombinant irisin in both in vitro and in vivo experiments.

As suggested, we have added these information in Method section (page 20).

We hope these experiments addressed the concern that was raised by the Reviewer.